A new comprehensive eye-tracking test battery concurrently evaluating the Pupil Labs glasses and the EyeLink 1000

http://orcid.org/0000-0002-6276-3332 Ehinger Benedikt V. 1 2 behinger@uos.de
http://orcid.org/0000-0002-3505-9999 Groß Katharina 1
Ibs Inga 1
http://orcid.org/0000-0003-3654-5267 König Peter 1 3
1 Institute of Cognitive Science, Osnabrück University , Osnabrück , Germany
2 Donders Institute for Brain, Cognition and Behaviour, Radboud University , Nijmegen , Netherlands
3 Department of Neurophysiology and Pathophysiology, University Medical Center Hamburg-Eppendorf , Hamburg , Germany
Negrello Mario
Electronic publication date: 2019 Jul 9
Publication date: 2019
Volume: 7
Electronic Location ID: e7086
Received 2019 Feb 5; Accepted 2019 May 7
Copyright: © 2019 Ehinger et al.
Copyright year: 2019
Copyright holder: Ehinger et al.
License: This is an open access article distributed under the terms of the Creative Commons Attribution License, which permits unrestricted use, distribution, reproduction and adaptation in any medium and for any purpose provided that it is properly attributed. For attribution, the original author(s), title, publication source (PeerJ) and either DOI or URL of the article must be cited.
License URL: https://creativecommons.org/licenses/by/4.0/

Keywords: Pupil dilation, Smooth pursuit, Microsaccades, Blinks, Eye-tracker benchmark, Accuracy and precision, Head movements, EyeLink 1000, Pupil Labs glasses, Calibration decay

Funding: European Union H2020-FETPROACT-2014, SEP-210141273, ID: 641321 socSMCs Deutsche Forschungsgemeinschaft (DFG) Open Access Publishing Fund of Osnabrück University This work was supported by the European Union (H2020-FETPROACT-2014, SEP-210141273, ID: 641321 socSMCs), the Deutsche Forschungsgemeinschaft (DFG) and the Open Access Publishing Fund of Osnabrück University. The funders had no role in study design, data collection and analysis, decision to publish, or preparation of the manuscript.

==============================
Eye-tracking experiments rely heavily on good data quality of eye-trackers. Unfortunately, it is often the case that only the spatial accuracy and precision values are available from the manufacturers. These two values alone are not sufficient to serve as a benchmark for an eye-tracker: Eye-tracking quality deteriorates during an experimental session due to head movements, changing illumination or calibration decay. Additionally, different experimental paradigms require the analysis of different types of eye movements; for instance, smooth pursuit movements, blinks or microsaccades, which themselves cannot readily be evaluated by using spatial accuracy or precision alone. To obtain a more comprehensive description of properties, we developed an extensive eye-tracking test battery. In 10 different tasks, we evaluated eye-tracking related measures such as: the decay of accuracy, fixation durations, pupil dilation, smooth pursuit movement, microsaccade classification, blink classification, or the influence of head motion. For some measures, true theoretical values exist. For others, a relative comparison to a reference eye-tracker is needed. Therefore, we collected our gaze data simultaneously from a remote EyeLink 1000 eye-tracker as the reference and compared it with the mobile Pupil Labs glasses. As expected, the average spatial accuracy of 0.57° for the EyeLink 1000 eye-tracker was better than the 0.82° for the Pupil Labs glasses (N = 15). Furthermore, we classified less fixations and shorter saccade durations for the Pupil Labs glasses. Similarly, we found fewer microsaccades using the Pupil Labs glasses. The accuracy over time decayed only slightly for the EyeLink 1000, but strongly for the Pupil Labs glasses. Finally, we observed that the measured pupil diameters differed between eye-trackers on the individual subject level but not on the group level. To conclude, our eye-tracking test battery offers 10 tasks that allow us to benchmark the many parameters of interest in stereotypical eye-tracking situations and addresses a common source of confounds in measurement errors (e.g., yaw and roll head movements). All recorded eye-tracking data (including Pupil Labs’ eye videos), the stimulus code for the test battery, and the modular analysis pipeline are freely available (https://github.com/behinger/etcomp).

Introduction

Eye-tracking has become a common method in cognitive neuroscience and is increasingly utilized by diagnostic medicine, performance monitoring, or consumer experience research (Duchowski, 2007; Holmqvist et al., 2011; Liversedge, Gilchrist & Everling, 2012). These applications are diverse, make use of many different eye movement parameters, and have different technical requirements. A single index might not be sufficient to characterize the suitability of an eye-tracker for all applications, but a more comprehensive test provides access to multiple indices for characterization (Hessels et al., 2015; Niehorster et al., 2018).

In the following, we will shortly highlight several of these eye movement parameters, their technical challenges and applications: accuracy is one of the most dominant and most reported characteristic of an eye-tracker. It is an index that correlates with eye-tracker performance in most tasks. During an experiment, accuracy can decay, for example, due to head movements, prompting researchers to recalibrate the eye-tracker multiple times during an experiment. A good accuracy is necessary in many applications, especially if fine differences in eye movements need to be resolved. Example applications can be found in saliency research (Itti, Koch & Niebur, 1998) or the reading literature (Rayner, 2009), where objects or words are close to each other. Precision refers to the variable error in the gaze coordinate signals. It is a measure of noisiness of an eye-tracker and consequently has influence on many paradigms. Especially for small eye movements like microsaccades (Rolfs, 2009) it is important to have a good (small) precision. Some eye-trackers are quite sensitive to head movements. This is especially important in populations that move a lot; for instance, infants or some clinical populations (Açık et al., 2010; Dowiasch et al., 2015; Cludius et al., 2017; Fischer et al., 2016). Similarly, head movements are to be expected in free moving mobile settings (Einhäuser et al., 2007, 2009; Schumann et al., 2008). In addition, in free moving mobile settings head movements are commonly accompanied by smooth pursuit (Marius’t Hart et al., 2009), yet another eye movement parameter. A different but very interesting eye behavior is blinks which can be related to dopamine levels (Riggs, Volkmann & Moore, 1981; but see Sescousse et al., 2018 for recent more nuanced evidence), saccadic suppression (Burr, 2005), or time perception (Terhune, Sullivan & Simola, 2016). Another eye-tracking parameter is pupil dilation, a physiological measure with many cognitive applications (Mathôt, 2018): It allows to track attention (Wahn et al., 2016), investigate decision making (Urai, Braun & Donner, 2018), and even communicate with locked-in syndrome patients (Stoll et al., 2013). These examples illustrate the diversity of eye movement parameters, but nevertheless only show a fraction. What becomes clear is that we need a large set of experimental tasks eliciting different eye movement types in a controlled manner in order to characterize an eye-tracker.

Estimating the performance of an eye-tracker is difficult, because many eye-tracking measures cannot be compared to a theoretical true value. For instance, standard calibration methods rely on participants accurately fixating visual stimuli, typically dots. However, even when participants think they fixate on a dot, their actual gaze point will never be perfectly resting on the dot. Unknown to them, miniature eye movements like drift and microsaccades move the gaze point around the fixation target (Rolfs, 2009). Nevertheless, to estimate the reliability of a single eye-tracker and compensate for the lack of ground truth at the same time, it is necessary to measure the participants’ gaze with two eye-trackers simultaneously: a top-of-the-line reference eye-tracker and the target eye-tracker (examples for this idea can be found in Titz, Scholz & Sedlmeier, 2018; Popelka et al., 2016; Drewes, Montagnini & Masson, 2011).

Consequently, we recorded the participants’ gaze with two video-based eye-trackers at the same time: the stationary EyeLink 1000 (SR research) and the mobile Pupil Labs glasses (Pupil Labs, Berlin, Germany). The EyeLink 1000 is a popular high-end, video-based, remote eye-tracker which we use as our reference. It is an eye-tracker with one, for a video based eye-tracker, of the best accuracy, and precision (Holmqvist, 2017) currently available. In principle, a dual-purkinje eye-tracker would be preferable due to the higher accuracy (Crane & Steele, 1985; Körding et al., 2001), but was not available. We chose to benchmark the mobile Pupil Labs eye-tracking glasses because they are special in several regards: For mobile eye-tracking glasses they offer high sampling rates (current versions 200 Hz per eye, our version up to 120 Hz per eye), the hardware and software is open source, and the eye-tracker is quite affordable. Depending on the specifications of the two eye-trackers, their prices can vary by a factor of 15. These features foster the wide usage of this mobile eye-tracker and motivate the comparison to the reference eye-tracker.

There is little published data on the performance of eye-trackers and even less independently from the manufacturers (Blignaut & Wium, 2014; Hessels et al., 2015; Niehorster et al., 2018). Worse, no standards to measure and report eye-tracker performance exist (Holmqvist, Nyström & Mulvey, 2012) and open source systematic benchmarks for eye-tracking devices are not available. However, as we have seen, the problem is complex, as single measures like spatial accuracy and precision, even though they are arguably two of the most useful single metrics, will never be able to fully describe the performance of an eye-tracker.

For these reasons, we developed a new paradigm to evaluate the data quality of the most common eye-tracking related parameters. Our test battery consists of: fixation and saccade properties in an artificial grid and in a free-viewing task, decay of accuracy (the tendency of observing worse accuracy over time, Nyström et al., 2013), smooth pursuit, pupil dilation, microsaccades, blink classification, and the influence of head motion.

To circumvent the need for theoretical true values, we make use of relative comparisons between two simultaneously recorded eye-trackers. Our large set of analyzed eye-tracking parameters offers a comprehensive characterization of the tested eye-trackers.

In order to make our analyses in this paper reproducible and to offer a dataset for benchmarking purposes, we made the recorded data (including the eye camera video streams) available on figshare (10.6084/m9.figshare.c.4379810). The source code of the eye-tracking test battery and the modular analysis pipeline are available on GitHub (https://github.com/behinger/etcomp).

Methods

Methods of data acquisition

Participants

We recruited 15 participants (mean age 24, range 19–28, nine female, zero left-handed, three left-dominant eye) at Osnabrück University. Eligibility criteria were: no glasses, no drug use, no photosensitive migraine or epilepsy, and more than 5 h of sleep the night prior to the experiment. A total of 11 additional participants were excluded from the analysis: six due to exceeding pre-specified calibration accuracy limits (two Pupil Labs glasses, three EyeLink 1000, and one both eye-trackers), and five due to software failures (see Table S1). Prior to the experiment, we used a calibrated online LogMar chart test (Open Optometry (2018); www.openoptometry.com) to ensure a visual acuity below 6/6 using a single test line with five letters. Ocular dominance was detected with the “hole-in-card” test by using the participants’ hands and centered gaze. After the experiment, we collected information about the participants’ age, gender, handedness, and eye color. We compensated the participants with either 9€ or one course credit per hour. The participants gave written consent and the study was approved by the ethic committee of Osnabrück University (4/71043.5).

Experimental setup and recording devices

The experiment was conducted at the Institute of Cognitive Science at Osnabrück University. In a separated recording room, we used a 24″ monitor (XL2420T; BenQ, Taoyuan City, Taiwan) with 1,920 × 1,080 pixels resolution and a 120 Hz refresh rate. The effective area of the monitor was 1,698 × 758 pixels because we displayed 16 visual markers for the Pupil Labs eye-tracker in the margins of the monitor (see Fig. 1). A single USB-loudspeaker was used to produce a beep sound for the auditory stimuli. The participants were seated at a distance of 60 cm to the monitor and the chamber light was kept on. We measured 52 cd/m2 from the point of view of the subject facing the monitor with the average gray luminance.

Figure 1 Experimental Setup. The remote eye-tracker EyeLink 1000 is located beneath the computer screen that displays the stimuli.

The participant wears the mobile Pupil Labs glasses. The auxiliary calibration monitor on the left was turned off during the experiment. (Photography Katharina Groß; subject’s consent to publish this image was granted).

The participants’ eye movements were recorded simultaneously by one stationary and one mobile eye-tracking device. A desktop mounted eye-tracker (EyeLink 1000; SR-Research Ltd., Mississauga, Ontario, Canada) was used to make monocular recordings of the participants’ dominant eye (500 Hz, head free-to-move mode). Concurrently, a mobile eye-tracker (Pupil Labs glasses; Pupil Labs, Berlin, Germany) was used to make binocular recordings of the participants’ eyes (Fig. 1). We specified the distance of camera to subject, angle and distance to monitor in the eye-tracker configuration files. We did not alter those settings if we had to slightly move the eye-tracker to adjust to an individual subject. The Pupil Labs glasses have three cameras: one world camera (1,920 × 1,080 pixels, 100° fisheye field of view, 60 Hz sampling frequency on a subset of 1,280 × 720 pixels) to record the participant’s view and one eye-camera for each eye (1,920 × 1,080 pixels, 120 Hz sampling frequency on a subset of 320 × 280 pixels). We recorded eye movements using Pupil Labs’ capture release 1.65 (November 2017).

We conducted the experiment using three computers: one stimulus computer and two recording computers, one for each eye-tracking device. To send experimental-messages (“triggers”) to the EyeLink 1000 recording computer we used the EyeLink Toolbox (Cornelissen, Peters & Palmer, 2002), for the Pupil Labs glasses we used zeroMQ packages (Wilmet, 2017). To temporally align the recordings during analysis, we used concurrent trigger signals via Ethernet at all experimental events. As we use two different protocols it is important to ensure that the time it takes to send a message and save it on the recording computer is very short. In separate measurements, we estimated round trip delay times with both recording computers of below one ms.

The experiment script was written in Matlab (R2016b; Mathworks, Natick, MA, USA) using the Psychophysics Toolbox 3 (Brainard, 1997; Pelli, 1997; Kleiner et al., 2007), EyeLink Toolbox (Cornelissen, Peters & Palmer, 2002), and custom scripts based on the ZMQ protocol for communication with the Pupil Labs glasses. The analyses were conducted using Python 3.5.2 (Van Rossum, 1995) with a version of Pupil Labs from April 2018 (git version: f32ef8e), pyEDFread (Wilming, 2015), NumPy (Oliphant, 2006), pandas (McKinney, 2010), and SciPy (Jones et al., 2001). For visualization, we used plotnine (Kibirige et al., 2018) and Matplotlib (Hunter, 2007).

Experimental design

All participants were recorded by a single, newly trained experimenter (Author I. I. with less than 1 year experience) under the supervision of an experienced experimenter (Author B. V. E. with more than 5 years experience).

The experiment lasted approximately 60 min. The session started with a brief oral explanation of the upcoming tasks, then we obtained written consent and an anamnesis questionnaire, which was used to exclude participants who suffer from a photosensitive migraine or epilepsy. We then identified their dominant eye and checked their acuity (see section “Participants” for procedures). Before the experiment, the experimenter emphasized the importance to look at the fixation targets.

The experiment consisted of six repetitions (blocks) of a set of 10 tasks (Fig. 2). Each block had the same order of tasks (see below). Participants read a written instruction prior to each task1 and saw a green fixation target at the center of the monitor. Participants then started the tasks at their own pace by pressing the space bar. In order to examine a variety of properties of the eye-trackers, each task either measures attributes of the eye-tracking devices (e.g., accuracy in a specific context which includes subject and experimenter), estimates suitability for specialist studies (e.g., pupil diameter and microsaccades), depicts a stereotypical eye-tracking situation (e.g., free viewing), or addresses aspects of more complex behavioral situations including head movements (e.g., yaw and roll head movements) and dynamic stimuli (e.g., smooth pursuit).

Figure 2 Experimental Design. Each block starts with calibration phase and is followed by a fixed sequence of the 10 tasks.

The experiment consisted of six identical blocks. Thus, each participant took part in six calibration procedures and a total of 60 tasks.

We kept the luminance of the desktop background and the room illumination constant at 52 cd/m2 during the whole experiment to prevent that the performances of the eye-trackers were affected by changes in ambient light intensity. Therefore, the calibration procedure and all tasks except the Pupil Dilation task (see section “Task 6: Pupil Dilation Task”) were presented using a gray background.

Methods of data analysis

For our analysis we built a flexible and modular pipeline that transforms raw eye-tracking data of two eye-trackers to dataframe-based data structures. One dataframe for the data samples, including timestamps, gaze points, velocities, pupil areas, and type (saccade, fixation, blink). One dataframe for the eye-tracking events, for example, fixations, saccades, blinks etc., and one dataframe for the experimental trigger messages which describe the conditions of the experiment. The pipeline is modularly programed and components can be easily exchanged. For example, it is easy to exchange the eye movement classification algorithms for event classification (blinks, saccades, and fixations). We hope this will improve the comparison of different algorithms in the future.

Preprocessing

A flowchart of the eye-tracking preprocessing pipeline is presented in Fig. 3. The raw EyeLink data (prefiltered using the EyeLink filter-option “extra”) already includes calibrated gaze, mapped to the monitor area and not much further pre-processing is needed. For Pupil Labs, we were forced to recalibrate the data, because online during recording, samples from the two eye cameras are not strictly interleaved in time and can confuse their calibration algorithm. We used the Pupil Labs’ Python API (Pupil Labs, 2018, git version: f32ef8e, April 2018) for recalibration and several of the following steps. Due to a (now resolved) bug in Pupil Labs’ software, we observed steep linear drifts between eye camera clocks and recording computer clock. Therefore, we recorded at every trigger message, both the current camera timestamp and the recording computer timestamp. Using linear regression, we could then synchronize the eye camera timestamps to the recording computer clock. Note that this step does not eliminate the inherent delay of 10 ms of the Pupil Labs’ cameras (personal communication with Pupil Labs).

Figure 3 Analysis flowchart. The flowchart illustrates the parallel steps from the recorded raw data to eye movement events (containing properties on fixation-, saccade-, and blink-events).

Because the Pupil Lab glasses is a mobile eye-tracker (head coordinate frame), but we compare it to a remote eye-tracker (world coordinate frame), we need to convert both eye-trackers to the same coordinate system. For this we chose spherical head coordinates. The EyeLink 1000 data are in world coordinates and will be transformed directly to spherical coordinates (see below). The Pupil Labs glasses are already in head centered coordinates, but are nevertheless first converted to screen (world) coordinates and then transformed back to the spherical head coordinate system. We cannot stay in the original head coordinate system because it has an arbitrary rotation compared to the final spherical coordinate system that both eye-trackers share.

In order to convert the Pupil Labs data from head coordinates to screen coordinates, we next needed to detect the display. For this we displayed 16 screen markers (in principle 4 would be enough, but we could not find a recommendation on how many should be used) in a 2.9° border at the edge of the monitor. These QR-like markers can be detected using the Pupil Labs’ API. A rectangular surface is then fitted to these markers and the calibrated gaze is mapped onto the surface using the Pupil Labs API. Only samples that are mapped to points inside the surface were considered in further analysis.

Next, for both eye-trackers, we converted the x (and y) gaze points of the raw samples from screen coordinates in pixels, to spherical angles in degree (with a reference system centered on the subject): βx = 2 · atan2(px · m, d), where βx denotes the azimuth angle (equivalent to the horizontal position) of the gaze points in visual degrees from the monitor center, px denotes the horizontal position relative to the center of the monitor in pixel, m denotes the unit conversion of pixel to mm of the monitor, and d denotes the distance to the monitor in mm. This new spherical coordinate system puts the subject at it’s origin. The radius of the sphere is the subject to monitor distance. The screen itself would be typically at 90° polar and 0° azimuthal, for convenience of plotting and interpretation, we label the screen’s center at 0°, 0° but perform all important calculations in the correct coordinate system (see section “Measures of Gaze Data Quality”).

We then detected and removed all bad samples that we did not consider in further analysis with the following exclusion criteria: no pupil detected, the gaze point was outside the monitor or the sample was marked as corrupt by the eye-tracker.

The experimental triggers that were sent from the stimulus computer to each of the recording computers were parsed into a pandas dataframe. Because recording computer clocks show drift over time relative to each other, we synchronized the timestamps of both eye-trackers by estimating the slope differences at the common event triggers. In addition, we corrected a 10 ms constant delay of the Pupil Labs glasses which compensated for their frame-capture delay (personal communication with Pupil Labs, verified using cross-correlation on two participants and visual inspection of overlaid signals).

Eye movement definition and classification

It is difficult to establish what an eye movement is, as the definition typically depends on the used algorithm, reference frame, and individual researcher (Hessels et al., 2018). Here, we focus on the comparison between devices, and an evaluation of algorithms defining fixations is beyond the scope of the present study. Therefore, we used identical algorithms for both eye-trackers wherever possible. Although, the further comparison of algorithms is outside of the scope of this paper, we want to highlight that our modular analysis pipeline greatly facilitates such comparisons.

In this article we define a saccade as a relatively fast movement in head-centered coordinates, classified by our algorithm (Engbert & Mergenthaler, 2006). Blinks are defined as reported by the eye-trackers (loss of sample (EyeLink) or confidence (Pupil Labs, Berlin, Germany)). Fixations are then defined as everything that is neither a saccade nor a blink. Thus, if the head rotates relative to the screen, but the world-centered direction of gaze stays constant, we would also count it as a fixation. Note that we do not have any moving objects and the participants’ heads were generally still. In the explicit head movement task, we only analyzed data after, but not during, the movement. One exception is the Smooth Pursuit task, were we explicitly analyze smooth pursuit, which, in contrast to our definition before, we define as the rotation of the eye in the direction and with comparable speed to the moving target.

Blink classification

Blinks are classified differently for the two eye-trackers. Pupil Labs’ blink classification algorithm depends on their confidence signal (see below), while Eye Link’s algorithm reports blinks when the pupil is missing for several samples. Therefore, it is not possible to use the same algorithm for both eye-trackers.

For the Pupil Labs data we used the Pupil Labs’ blink classification algorithm with minor adjustments. Pupil Labs classifies blinks based on time-smoothed confidence values of the samples which (in the version used in this paper) reflects the ratio of border pixels of the thresholded pupil overlap with a fitted ellipse. The Pupil Labs’ blink classification algorithm uses a thresholded smoothed differential filter-output to classify large changes in confidence and thereby identifies the start and the end of blinks. We noticed that the blink classification algorithm sometimes reported very long blinks (20 s or longer) and added a criterion that a blink can only have a start time point if it also has an end time point. Our code-change was that in case we found multiple consecutive blink start point candidates, we only used the last one. For the EyeLink data, we used the blinks that were already classified by its proprietary algorithm during recording.

For the subsequent saccade classification, we regarded the samples ±100 ms around a reported blink event as additional blink samples (Costela et al., 2014) and accounted for them during saccade classification. For the task analyses which rely directly on sample data, we excluded all blink samples.

Engbert and Mergenthaler Saccade classification

We used the velocity based saccade classification algorithm proposed by Engbert & Kliegl (2003) and Engbert & Mergenthaler (2006) in the implementation by Knapen (2016). Velocity based saccade classification algorithms use the velocity profile of eye movements to extract saccade intervals. The algorithm was originally developed to identify microsaccades, but by adjusting the hyperparameter (λ), it can be used for general saccade classification (for more details see Engbert & Mergenthaler, 2006).

The implementation we used requires a constant sampling rate, and we first interpolated the samples recorded by the Pupil Labs glasses with piecewise cubic hermite interpolating polynomials to obtain samples at a sampling rate of 240 Hz. Subsequently, the classified saccade timings were applied to the individual (non-interpolated) samples. We did not interpolate the EyeLink data samples as the sampling rate is constant at 500 Hz or constant at 250 Hz. For all saccade classifications we used a λ of 5.

Classification of fixations

We labeled all samples as fixation samples that were neither classified as blink nor saccade samples. We removed all fixation events shorter than 50 ms.

Notes on sampling frequencies

Fast sampling rates are important to shorten online delay for gaze-dependent experiments (e.g. Ehinger, Kaufhold & König, 2018) and increase accuracy of event onsets for example, EEG/ET co-registered experiments (Dimigen et al., 2011; Ehinger, König & Ossandón, 2015). In this paragraph we analyse the reported and effective sampling rates of both eye-trackers.

The EyeLink 1000 was sampled monocularly with 500 Hz for 10 participants and due to a programming mistake with 250 Hz for the other five participants. The recorded samples show that the empirical sampling rate matches the theoretical one perfectly.

Both Pupil Labs’ eye cameras sampled each with 120 Hz. Our empirical eye camera wise inter-sample distances confirm the theoretic sampling rates of 120 Hz. After the fusion and mapping to gaze-coordinates, Pupil Labs, as maybe expected, reports a sampling rate of 240 Hz. But this is not the effective sampling rate: The eye cameras are not synchronized to sample in anti-phase to each other (see Fig. 4). In our data, we found a uniform phase relation, indicating that participants’ effective sampling rates range from close to 120 Hz to close to 240 Hz (see Fig. 4B).

Figure 4 (A) Pupil Labs binocular recording.

Two cameras take samples of the eyes. Each has a fixed (and reliable) sampling rate of 120 Hz. During startup, the relative phase of the sampling timepoints of the two cameras is random. If we use the Pupil Labs fusion algorithm (green samples), which pairwise uses the eye-cameras’ samples, we will always get a steady sampling rate of 240 Hz regardless of the actual information content. (B) Using the eye-camera timestamps we calculate inter-sample time distances (shown also in A). Perfect anti-phasic behavior should show as a cluster around the 240 Hz line, perfect phasic behavior as a cluster around 120 Hz. Mixed phase seems to be the rule. (C) The consequence of a bad eye fusion algorithm. Inline with the temporal averaging shown in (A) the gaze position is also linearly interpolated. Nevertheless, we often observed staircase like patterns (see also section “Results Task 3: Free Viewing”). We think this is due to the 4D binocular calibration function that does not take time-delays into account during the fit.

In addition, we found two types of artifacts. One is visible in Fig. 5, which occurred for some subjects and has an unknown origin. Another (possibly related) artifact has a stereotypical step-function appearance which is especially visible during saccades (see Fig. 4). Both artifacts are likely problematic for the velocity-based saccade classification algorithm. For the latter, we offer an explanation of possible origin: during calibration, a 4D to 2D polynomial regression function is fitted. In order to do so, pairs of eye-coordinates (x-y from both cameras, making up the 4D vector) are mapped to the coordinates of a reference point of the world camera. This is done by finding the individual eyes’ sample that is closest in time to the target sample. This calibration fit will implicitly compensate for the delay of the two eye signals (except in the case of an in-phase relation). This in itself is sub-optimal (as samples of two different time points and thus eye positions are combined), but not alone the cause of the artifact. During gaze production, that is the application of the fitted polynomial function, samples are combined in a alternating fashion (Fig. 4A, green dots). The resulting time sample is always the average between the alternating eye samples and thus, as discussed before, has a perfect 240 Hz temporal distance. This effectively corrects again for the time-difference between eye camera samples, thereby introducing the step-like artifact. Disclaimer: we tried to be thorough in our investigation, but we are still unsure of the source of the artifact. It is certainly possible that other factors play a role and further simulations should be undertaken to pin down the exact source. We think eliminating this artifact could noticeably improve the performance of the Pupil Labs eye-tracker in the binocular recording condition, but this is outside the scope of the present paper.

Figure 5 Annotated samples from the accuracy task. Fixations: green, saccades: dark, for (A) a good subject and (B) a subject with Pupil Labs artifacts.

Measures of gaze data quality

Spatial accuracy in visual angle

The spatial accuracy of an eye-tracker refers to the distance of the measured gaze point and the instructed target (Holmqvist, Nyström & Mulvey, 2012). We calculated this angular difference by the cosine distance between two vectors: the mean gaze point (f=(fxfy)) and target location (t=(txty)). For this calculation, we converted the vectors from the Spherical coordinate system to the Cartesian one, which allows us to use the formula for the cosine distance: θ=acos(f⋅t‖f‖‖t‖). After conversion from radians to degrees, this results in the angular difference between 0 and 180°. For the conversion from spherical coordinates to cartesian, we rotated the polar and azimuthal angle by 90° so that the center of the screen is not at <0°, 0°, 60 cm> but at <90°, 90°, 60 cm> and consequently differences in both polar and azimuthal angle influence the angular distance equivalently.

During the calibration procedure, the distance between subsequent dots might be larger. Participants typically make catch-up saccades for saccades with large amplitude and small eye movements during fixation periods. Therefore, the gaze data might contain several candidate fixations for analysis. Holmqvist (2017) showed that the selection procedure is uncritical and we decided to use the last ongoing fixation, right before the participants confirmed fixation by pressing the space bar.

Our reported aggregate measure of accuracy is the 20% winsorized mean (Wilcox, 2012) spherical angle between the displayed target and the estimated participant’s fixation location.

Spatial precision

Spatial precision refers to the variable error in the gaze coordinate signals; an estimate of the variance of the noise. A good precision is reflected by a small dispersion of samples, as the distances between the samples are small when the samples are close to each other. We make use of the two most popular spatial precision measures, root mean squared (RMS) and the standard deviation.

The proximity of consecutive samples is assessed with the RMS of inter-sample distances: Let d((xiyi),(xi−1yi−1)) denote the angular distance (see section “Measures of Gaze Data Quality”) between sample (xiyi) and (xi−1yi−1). Precision was calculated as:

θRMS=1n∑i=1nd((xiyi),(xi−1yi−1))2

The spatial spread is assessed with the standard deviation of the sample locations. The standard deviation for a set of n data samples is calculated as: Let d((xiyi),(x¯y¯)) denote the angular distances between the mean fixation location (x¯y¯) (with x^=1n∑xi) and each fixation sample.

θsd=1n∑i=1nd((xiyi),(x¯y¯))2

We report fixation spread measured by 20% winsorized average values of standard deviation or inter-sample-distance measured by RMS.

Pupil dilation

In the Pupil Dilation task, we measure the pupil size of the participants as a reaction to different luminance stimuli. Measuring the pupil size can be done by means of diameter (Pupil Labs glasses) or pupil area (EyeLink 1000). With the Pupil Labs glasses, pupil diameter is estimated from a fitted ellipse. With the EyeLink 1000, pupil area is calculated as the sum of the number of pixels inside the detected pupil contour. We converted the pupil diameters reported by Pupil Labs into pupil area using: A=14π⋅l1⋅l2 where A denotes the ellipsis area, l1 denotes the semi-major axis and l2 denotes the semi-minor axis. In our experiment, pupil area is reported in pixels or arbitrary unit. The absolute pupil size is not important for the current study and due to lacking pupil calibration data, a conversion to is not possible. Pupil size fluctuates globally over blocks due to attention or alertness. We normalized the pupil area to the median of a baseline period (see section “Task 6: Pupil Dilation Task”).

Tasks

Task sequence

At the beginning of each block, directly after the eye-tracker calibration, we presented a grid task, that was designed to assess the spatial accuracy of the eye-trackers. In addition, we used the grid task right before and after a controlled block of head movements. Furthermore, we placed the fixation heavy tasks (Microsaccade and Pupil Dilation) in between tasks which were more relaxing for the participants (Blinks and Free viewing Accuracy).

Fixation targets

Throughout the experiment, we used three different fixation targets: for manufacturer calibration/validation, we used concentric circles following the Pupil Labs specifications in order to detect reference points from the world camera. For most fixation tasks we used a fixation cross that was shown to reduce miniature eye movements (Thaler et al., 2013). For several tasks, we used a bullseye (outer circle: black, diameter 0.5°, inner circle: white, diameter 0.25°): firstly, for smooth pursuit because diagonal fixation dot movement looked better aesthetically. Secondly, for microsaccades, as we did not want to minimze microsaccades. Thirdly, for pupil dilation we used the bullseye because it is visible regardless of background illumination.

Calibration

We calibrated the devices at the beginning of each block using a 13-point randomized calibration procedure. We used concentric rings as fixation points which can be detected by the Pupil Labs glasses’ world camera. The 13 calibration points were selected as a subset of the large grid from the accuracy task (see section “Task 1/Task 7/Task 10: Accuracy Task with the Large and the Small Grid”). Calibration points were manually advanced by the experimenter. An automatic procedure (EyeLink default setting) was not possible, because the calibration of both recording devices was performed at the same time. After calibration, a 13-point verification was performed which was identical in procedure but with a new sequence. The accuracies were calculated online by both devices. The devices were recalibrated if necessary, until the mean validation-accuracies met the recommendations by the manufacturers. The mean validation-accuracy limit for the EyeLink 1000 was 0.5° where the validation-accuracy of each point was not allowed to exceed 1° (SR-Research manual). The mean validation-accuracy limit for the Pupil Labs glasses was 1.5° (personal communications with Pupil Labs). If more than 10 unsuccessful calibration attempts were made, with adjustments of the eye-trackers in between, we stopped the recording session and excluded the participant from the experiment.

Task 1/Task 7/Task 10: accuracy task with the large and the small grid

We used a fixation grid to evaluate the difference between the location of a displayed target and the estimated gaze point. We estimated absolute spatial accuracy and in addition, decay of the calibration accuracy over time. We used two variants of the accuracy task, a large grid based on a 7 × 7 grid and a small grid based on a subset of 13 points. The large grid accuracy task is shown directly after the initial calibration of each block. This allowed us to estimate the accuracy of the eye-trackers with almost no temporal decay. To additionally investigate the decay of the calibration, we recorded the small grid tasks after the participant completed five different tasks (after about 2/3 of the block ≈4 min 42 s) and after two further tasks involving head movements (≈6 min 18 s).

Task with the large grid

The participants were instructed to fixate targets that appeared at one of the 49 crossing points of a 7 × 7 grid. The crossing points were equally spaced in a range from −7.7 to 7.7° vertically and −18.2 to 18.2° horizontally. At each crossing point a target appeared once, so in total 49 targets were shown during every task repetition. The participants were asked to saccade to the target and fixate it, and once they felt their eyes stopped moving, to press the space bar to continue. The center point was used as the start and end point.

A sample screen is visible in Fig. 1 and an animated gif is available on GitHub (https://github.com/behinger/etcomp/tree/master/resources).

Task with the small grid

The small grid task is analogous to the large grid task, but with a subset of 13 target points. These points were also used in the calibration procedure and spanned the whole screen.

Randomization of the large grid

A naive approach of randomization of the sequence of fixation points would lead to heavily skewed distributions of saccade amplitudes. Therefore, we used a constrained randomization procedure to expose participants to as-uniform-as-possible saccade amplitudes and angles distributions. We used a brute-force approach maximizing the entropy of the saccade amplitude histogram (17 × 1 degree bins) and the saccade angle histogram (10 × 36 degree bins) with an effective weighting (due to different bin widths) of 55:45%. This allowed for better subject comparisons as the between-subject-variance due to different saccade parameters is minimized with this procedure.

Randomization of the small grid

The sequence of the target positions was naively randomized within each block and for each participant.

Measures of the large grid

For the large grid we evaluated how accurate the participants fixated each target, that is the offset between the displayed target and the mean gaze position of the last fixation before the new target is shown (see section “Measures of Gaze Data Quality”). Furthermore, we analyzed the precision of the fixation events by evaluating the RMS and SD (see section “Measures of Gaze Data Quality”).

Measures over all grid tasks

Because we recorded grid tasks at several time points during a block, we were able to obtain accuracy measures with no decay (directly after initial calibration), after some temporal drift (2/3 of the block elapsed), and after provoked head movements (yaw and roll task 2.3.10). The accuracy decay over time showed effects for which statistical significance could not directly be seen. Therefore, after plotting the data, we decided to use a robust linear mixed effects model with conservative Walds t-test p-value calculation (df = Nsubjects−1). We used the robust version as we found out (after inspecting the data) that there are outliers at all levels, single element, blocks, and subjects. These are accommodated by the winsorized means in the general analysis, but not if we would have performed a normal linear mixed model (LMM). For this we defined the LMM accuracy ∼ 1 + et * session (1 + et * session | subject \ block) and evaluated it with the robustlmm R package (Koller, 2016). The maximal LMM containing all random slopes did not converge and therefore we used the simplified model as stated above.

Task 2: smooth pursuit

Smooth pursuit (defined as slow eye movements relative to saccades) is a common eye movement that occurs when the occulomotor system tracks a moving object. It is especially common while we move relative to a fixated object and, therefore, elemental to classify reliably for mobile settings.

Task

To analyze smooth pursuit movements, we followed Liston & Stone (2014) and adapted their variant of the step-ramp smooth pursuit paradigm. The participants fixated a central target and were instructed to press the space bar to start a trial. In this task we used a bullseye fixation target. The probe started after a random delay. The delay was sampled from an exponential function with a mean of 0.5 s with a constant offset of 0.2 s and truncated at 5 s. This results in a constant hazard function and counteracts expectations of motion onset (Baumeister & Joubert, 1969). The stimuli were moving on linear trajectories at one of five different speeds (16, 18, 20, 22, 24°/s). The trial ended once the target was at a distance of 10° from the center. We used 24 different orientations for the trajectories spanning 360°. To minimize the chance of catch-up saccades, we chose the starting point for each stimulus such that it took 0.2 for the target to move from the starting point to the center. We instructed the participants to follow the target with their eyes as long as possible.

Randomization

One block consisted of 20 trials with a total of 120 trials over the experiment. Each participant was presented with each of the 120 possible combinations of speed and angle once, randomized over the whole experiment.

Measures

Automatic smooth pursuit classification is still in its infancy (but see Larsson et al., 2015; Pekkanen & Lappi, 2017) and, therefore, we opted to use a parametric smooth pursuit task that can be evaluated with a formal model. To analyze smooth pursuit onsets and velocities we generalized the model used by Liston & Stone (2014) to a Bayesian model. First, we rotated the x-y gaze coordinates of each trial in the direction of the smooth-pursuit target. Now an increase in the first dimension is an increase along the smooth-pursuit target direction. We then restricted our data fit to samples up to the first saccade exceeding 1° (a catch-up saccade) or up to 600 ms after trial onset. We used the probabilistic programming language STAN to implement a restricted piece-wise linear regression with two pieces. The independent variable of the regression is the eye position along the smooth pursuit trajectory which should be a positive component (else the eye would move in the opposite direction to the smooth pursuit target). The first linear piece is constrained to a slope of 0 and a normal prior for the intercept with mean 0 and SD of 1° (in the rotated coordinate system). The hinge or change-point has a prior of 185 ms post-stimulus onset with a SD of 300 ms. The slope of the second linear piece is constrained to be positive and follows a 0-truncated normal distribution with mean 0 and SD of 20°/s. The noise is assumed to be normal with a prior SD of 5°. For the hinge we used a logistic transfer function to allow for gradient-based methods to fit the data. We want to note that this analysis is sensitive to classifying the initial saccade correctly and does not distinguish between catch-up saccades and initial reaction saccades. For this paper, we assume that the impact of these inadequacies can be compensated by the robust winsorized means that we employ at various aggregation levels. For each trial we take the mean posterior value of the hinge-point and the velocity parameter and use winsorized means over blocks and subjects to arrive at our group-level result. In addition, we count the reported number of saccades during the movement of the target.

Task 3: free viewing

Task

For the free viewing task, we presented photos of natural images consisting mostly of patterns taken from Backhaus (2016). Participants fixated on a central fixation cross for on average 0.9 s with a uniform random jitter of 0.2 s prior to the image onset. The participants were instructed to freely explore the images. During each of the six blocks, we showed three images (900 × 720 pixels) for 6 s, thus 18 different images in total.

Randomization

The order of the 18 images was randomized over the experiment and each image was shown once. Due to a programming mistake, the first participant saw five different images compared with the other participants. These deviant images were removed from further analysis.

Measures

We compared the number of fixations, fixation durations, and saccadic amplitudes between eye-trackers. Furthermore, we visually compared the gaze trajectories of the two eye-trackers to get an impression of the real world effects of spatial inaccuracies. We excluded the first fixation on the fixation cross. For the central fixation bias we smoothed a pixel-wise 2D histogram with a Gaussian kernel (SD = 3°).

Task 4: microsaccades

Task

In order to elicit microsaccades, we showed a central fixation target for 20 s. The participants were instructed to continue the fixation until the target disappeared. In this task, we used the bullseye fixation target and for obvious reasons, not the fixation target that minimizes microsaccades (Thaler et al., 2013).

Measures

We evaluated the number of microsaccades, the amplitudes of the microsaccades, and the form of the main sequence. For this task, we ran the Engbert & Mergenthaler (2006) algorithm only on this subset of data specifically for each block.

Task 5: blink task

Task

The participants fixated a central fixation target and were instructed to blink each time they heard a beep. The 300 Hz beep sound chimed 100 ms for seven times with a pause of 1.5 s between every beep. Each sound onset was uniformly jittered by ±0.2 s in order to make the onsets less predictable for the subjects. We used the Psychophysics Toolbox’s MakeBeep function to generate the sound.

Measures

We evaluated the number of reported blinks and blink durations. Note that different blink classification algorithms were used (see section “Eye Movement Definition and Classification”).

Task 6: pupil dilation task

Task

In this task, we varied the light intensity of the monitor to stimulate a change of pupil size. During the entire task, a central fixation target was displayed which the participants were instructed to fixate. Each block consisted of four different monitor luminances (12.6, 47.8, 113.7, and 226.0 cd/m2) corresponding to 25%, 50%, 75%, and 100%. Before each target luminance, we first showed 7 s (jittered by ±0.25 s) of black luminance (0.5 cd/m2, 0%). This was done in order to allow the pupil to converge to its largest size. Then, one of the four target luminances was displayed for 3 s (jittered by ±0.25 s).

Randomization

The order of the four bright stimuli was randomized within each block.

Measures

We analyzed the relative pupil areas per luminance. We first converted the Pupil Labs pupil signal from diameter to area (see section “Measures of Gaze Data Quality”). Then we calculated the normalized pupil response by dividing through the median baseline pupil size 1 prior to the bright stimulus onset. We did this as visual inspection of raw traces showed that in many trials the 7 s black luminance was not sufficient to get back to a constant baseline and in other trials the pupil seemed converged, but not on the same baseline level indicating either block-wise attentional processes, different distance of eye camera to eyes, or other influences. The normalized pupil area is therefore reported in percent area change to median baseline.

Task 8/9: head movements

Task roll movement

In this task, we examined the gaze data while the participants tilted their heads. The participants saw a single rotated line in each trial. In each trial the line was presented at seven different orientations (−15, −10, −5, 0 (horizontal), 5, 10, 15). The participants were instructed to rotate their head so that their eyes are in line with the line on the screen, while fixating the target. Once the participants aligned their eyes with the line, they pressed the space bar to confirm the fixation/position and the next line was shown.

Randomization for roll movement

The sequence of the lines was randomized within each block and for all participants. The order of the roll and yaw tasks alternated in each block for a participant. Half of the participants started with the roll task, the other half with the yaw task.

Measures for roll movement

Because the subjects continued to fixate on the fixation cross at the center of the line and rolled their head, often no new fixation was classified. Therefore, we analyzed the winsorized average fixation position 0.5 before the button press.

Task yaw movement

In this task the participant performed 15 yaw movements during one block. For this purpose, we showed targets at five equally spaced positions on a horizontal line (Positions: −32.8, −16.7, 0, 16.7, 32.8). The participants were instructed to rotate their head so that their nose points to the target and then fixate it. Once they fixated the target, they pressed the space bar to confirm the fixation and the next target appeared.

Randomization for yaw movement

The positions of the 15 targets were randomized within one block.

Measures for yaw movement

We analyzed the accuracy of the estimated gaze point of the participant on the last fixation before subjects confirmed the yaw movement.

Results

We recorded the eye gaze position and pupil diameter of 15 participants concurrently with two eye-trackers. In Fig. 5, we show exemplary traces of a single participant for both eye-trackers. We see an overall high congruence of the recorded samples. Often even small corrective saccades seem to match between the two eye-trackers. But of course, important information which cannot be observed visually is hidden in the traces and requires quantitative analyses.

Note that for the following results we generally first calculated the winsorized mean for each participant over blocks and then report a second winsorized mean and the inter-quartile range (IQR) over the already averaged values. In other words, we report the IQR of means, not the mean IQR.

Results: calibration

In the great majority of eye-tracking experiments, eye-trackers first have to be calibrated. That is, (typically) a mapping from a pupil position coordinate frame to a world coordinate frame needs to be estimated. We used an experimenter-paced 13-point calibration procedure to calibrate both eye-trackers simultaneously. We made use of the eye-trackers’ internal validation methods.

For the EyeLink data, the winsorized mean validation-accuracy was 0.35 (IQR: 0.31–0.38), for Pupil Labs it was 1.04 (IQR: 0.96–1.14). These results are certainly biased as a selection bias was introduced when we repeated the calibration if the validation-accuracy was worse than our prespecified validation-accuracy limits (0.5 for EyeLink 1000 and 1.5 for Pupil Labs glasses). Besides the participants that were completely excluded from further analysis (see section “Participants”), only for seven validations (of in total 6 × 15 × 2 = 180 eye-tracker validations) a validation below the limits was not possible (see Figs. 6C and 6D). Note that these seven validations are equally spread over eye-trackers and are uncorrelated over eye-trackers/sessions. For unknown reasons, the Pupil Labs validation data was not saved for three participants.

Figure 6 (A) Calibration Validation display.

(B) A 13-point calibration procedure paced by the experimenter was performed at the beginning of each block. During calibration the built-in procedure of each eye-tracker was used. Both eye-trackers were calibrated simultaneously. (C) Reported 13-point validation-accuracy of the eye-trackers’ built-in procedures with winsorized mean and 95% winsorized mean confidence intervals. Note that we show disaggregate data over participants and report mean and CI over blocks instead. The values aggregated over participants first are reported in the text. (D) Reported 13-point validation-accuracy of the eye-trackers’ built-in procedures split over participants (same data as in C). Each point indicates the accuracy value for one participant in one block. Calibration accuracy data of Pupil Labs was missing for three participants. The prespecified accuracy limits (see section “Experimental Design”) were exceeded in only 7 out of 180 validations without resulting in a recalibration.

In summary, we succeeded in calibrating both eye-trackers simultaneously in the validation-accuracy ranges that are recommended by the eye-tracker manufacturers.

Results Task 1/7/10: accuracy task with small grid I and II

Spatial accuracy and precision are the most common benchmark parameters of eye-trackers. We measured those by asking participants to fixate points on a 49 point fixation grid. We report 20%-winsorized means, first aggregated over the 49 grid points, then over the six blocks and finally over the 15 participants (Fig. 7).

Figure 7 (A) Accuracy task.

(B) The participants fixated single points from a 7 × 7 grid and continued to the next target self-paced by pressing the space bar. The fixation while pressing the space bar was used for analysis. (C) Kernel densities of fixation durations. The thick line indicates the average over all data points irrespective of subjects. (D) Spatial accuracy: 20% winsorized mean and between subject 95% winsorized confidence intervals are shown. Green lines show 20% winsorized means over six blocks of individual subject, where each block was calculated by the 20% winsorized mean accuracy over 49 grid points. (E) 2D-Distribution of fixations around the respective grid points. 95% bivariate t-distribution contours (df = 5) are shown. That is, a robust estimate where 95% of a grid points’ fixations are expected to fall. (F) Spatial accuracy over the time of one block. Dashed line shows average at the first measurement point facilitating comparison to the later two measurement points. (G) Difference of actual fixation position and fixation target position. Bivariate t-distribution contours (df = 5) over all fixations over all participants. (H) Precision: root means squared (RMS) inter-sample distance. (I) Precision: fixation spread (SD). (J) SD over grid point positions. (K) Pupil Labs—EyeLink fixation duration difference.

The winsorized mean accuracy of EyeLink was 0.57° (IQR: 0.53–0.61°), of Pupil Labs 0.82° (IQR: 0.75–0.89°), with a paired difference of −0.25° (CI95 [−0.2 to −0.33°]). Therefore, Pupil Labs has in this condition a ≈45% worse spatial accuracy value than EyeLink. These accuracies have to be taken as best-case accuracies as they were measured shortly after the calibration procedure.

We quantified the spatial precision using the inter-sample distances (RMS) and the fixation spread (standard deviation). For EyeLink the winsorized mean RMS was 0.023° (IQR: 0.014–0.04°), for Pupil Labs 0.119° (IQR: 0.096–0.143°)2 , with a paired difference of −0.094° (CI95 [−0.077 to −0.116°]). Therefore, Pupil Labs has a ≈500% worse RMS precision than EyeLink. Note that we used the maximal filter settings for EyeLink, this will reduce the RMS, whereas no such option exists in Pupil Labs. We expect the binocular fusion issues and the differing sampling rates (see Fig. 4) to inflate this measures. The interaction between RMS and sampling rate is complex, as in principle both, increased RMS due to higher sampling rate (because more noise is included; compare Holmqvist et al., 2011) and reduced RMS due to higher sampling rate (because of quadratic summation) are possible. Given recent findings (Holmqvist, Zemblys & Beelders, 2017), the former seems more likely than the latter.

The arguably more intuitive spatial precision measure is standard deviation as it gives an intuitive measure of fixation spread. For EyeLink, the winsorized mean standard deviation was 0.193° (IQR: 0.164–0.22°), for Pupil Labs 0.311° (IQR: 0.266–0.361°), with a paired difference of −0.118° (CI95 [−0.073 to −0.174°]). Here, similar to accuracy, Pupil Labs shows a ≈50% worse precision than EyeLink.

We measured a subset of grid points at three points during a block: immediately after calibration, after 279 s (95-percentile: 206 s–401 s), and after 375 s (95-percentile: 258 s–551 s). Because differences are not as evident as in other conditions, a robust LMM was used to estimate the decay in accuracy over time. EyeLink showed a quite stable calibration accuracy. At the second measurement, average accuracy was worse than initial measurement by 0.06° (t(14) = 3.86, p = 0.002), at the third measurement, only marginally worse than initial measurement by 0.03° (t(14) = 2.1, p = 0.05). In contrast, Pupil Labs showed a much stronger decay. At the second measurement the accuracy dropped already by 0.25° (t(14) = 11.27, p < 0.001) to ≈1.1°. Interestingly, even after head motions, the accuracy did not get much worse with a difference to the initial measurement of 0.29° (t(14) = 13.07, p < 0.001).

For EyeLink, we estimated an winsorized average fixation duration on one grid point of 1.03 s (IQR: 0.82–1.28 s), for Pupil Labs 1.09 s (IQR: 0.89–1.34 s), with a paired difference of −0.07 s (CI95 [−0.06 to −0.08 s]). As clearly evident in Fig. 7K, there are two sources for the observed difference. For one, Pupil Labs often misses catch-up saccades, thereby prolonging average fixation duration. On the other hand the initial peak around 0 is positively biased, indicating that also for other fixations, Pupil Labs offers longer fixation durations. This might be a consequence of our use of the sample-wise saccade classification algorithm.

In conclusion, we found that EyeLink, as well as Pupil Labs, showed rather good spatial accuracies and precision values. As expected from the reference eye-tracker, EyeLink exhibited better performance. A decay of calibration was found only for Pupil Labs, where the calibration decayed by ≈30% after 4 min 30 s. It is therefore important to recalibrate the Pupil Labs glases more often to keep the same level of accuracy and spatial precision as initially after calibration.

Results task 2: smooth pursuit

To elicit and measure smooth pursuit, we implemented a smooth pursuit test battery proposed by Liston & Stone (2014), with a target moving from the center of the screen outward using 24 different angles and five different speeds (Fig. 8). We developed and fitted a single-trial Bayesian model to estimate the tracking onset and the tracking velocity (see section “Task 2: Smooth Pursuit”).

Figure 8 (A) Smooth Pursuit task.

(B) The participants made smooth pursuit eye movements imposed by a step ramp paradigm (see section “Task 2: Smooth Pursuit”). (C) Analysis model: single-trial Bayesian estimates of a hinge-regression model. The main parameters were the offset of the initial fixation, the tracking onset of the smooth pursuit eye movement and the tracked velocity (slope). Prior to modelfit we rotated the data to align all tracking target directions. (D) Example model fit: one trial of one participant. We used the data up to a first possible catch-up saccade (green dots). Uncertainty in model fit is visualized by plotting 100 random draws from the posterior. Red dots (overlapping for both eye-trackers) indicate estimated smooth pursuit onset. (E) Winsorized average tracking onset for each participant. (F) Winsorized average tracking velocities for each participant. (G) Amplitudes of catchup saccades. Pupil Labs reports smaller catchup saccade amplitudes independently of target velocity.

For EyeLink the winsorized mean smooth pursuit onset latency was 0.241 s (IQR: 0.232–0.250 s), for Pupil Labs 0.245 s (IQR: 0.232–0.252 s). The estimated onset latencies were equal between eye-trackers with an average difference of −0.001 s (CI95 [0.003 to −0.007 s]). Our analysis method estimates the onset latency using many samples before and after the onset. This could hide potential latency effects without such a structural analysis method.

For EyeLink the winsorized mean tracking velocity was 10.5°/s (IQR: 8.5–12.52°/s), for Pupil Labs 13.1°/s (IQR: 11.7–14.8°/s), with a paired difference of −2.4°/s (CI95 [−1.5 to −4.0°/s]). These pursuit velocities are much smaller than the target velocities (but accurately estimated, for example see Fig. 8D). These slow pursuit velocities are accompanied by a high frequency of catch-up saccades. Specifically, the distance the target is tracked is covered evenly by pursuit movements and catch-up saccades. In addition to the large number of catchup saccades, we observed that Pupil Labs reported smaller catch-up saccade amplitudes, independently of the target velocity (Fig. 8G). If we take the lower sampling rate of the Pupil Labs eye-tracker into account, we see that each catch-up saccade consists of fewer samples (compared to the EyeLink). If we have fewer samples, reported saccades will also exhibit smaller amplitudes (similar to Fig. 9F). Consequently, tracking velocities are also biased, as samples later in time (and thus with higher eccentricity) are included in the model for Pupil Labs compared to EyeLink. This could explain the bias of the model to fit steeper slopes in Pupil Labs compared to EyeLink.

Figure 9 (A) Free Viewing task.

(B) The participants freely explored the images for 6 s. (C/D) Scanpaths from one participant (EyeLink: blue; Pupil Labs: orange; fixation samples: brighter color; saccade samples: darker color). (E) Heatmaps for EyeLink and Pupil Labs on the base of reported fixations with a Gaussian kernel with 3° smoothing. (F) Histogram of saccade amplitude. Binwidth of 0.25°. (G) Histogram of fixation duration. Binwidth of 25 ms. (H) Winsorized mean number of fixations per image.

In summary, smooth pursuit signals could be detected by both eye-trackers. There were large biases between eye-trackers, even though the artificial task structure should make smooth pursuit detection easy.

Results task 3: free viewing

We presented a total of 18 images in the unrestricted Free-Viewing task. The images were displayed for 6 s each and showed mostly natural patterns and textures, and scenarios (Fig. 9).

For EyeLink, the winsorized mean fixation count was 17.2 (IQR: 16.2–18.3), for Pupil Labs 14.1 (IQR: 12.7–15.6). Thus, Pupil Labs reported on average 2.5 (CI95 [3.8–1.7]) fewer fixations per 6 s. For EyeLink, the winsorized mean fixation duration was 0.271 s (IQR: 0.246–0.30 s), for Pupil Labs 0.330 s (IQR: 0.310–0.352 s), with a paired difference of −0.054 s (CI95 [−0.039 to −0.072 s]). For EyeLink, the winsorized mean amplitude winsorized mean was 4.24° (IQR: 3.63–4.89°), for Pupil Labs 3.69° (IQR: 3.15–4.28°), with a paired difference of 0.39° (CI95 [0.69–0.09°]).

As shown in Fig. 9E, we find the classical central fixation bias (compare Tatler, 2007). In Figs. 9C and 9D we show the scan-paths of one participant during the Free Viewing task. The recorded scan-paths from EyeLink and Pupil Labs differ noticeably. Locally, the Pupil Labs data show a lower sampling frequency and alternating gaze position (indicating poor fusion of the two eyes’ data) resulting in high variance of eye position, especially visible during saccades. Globally, if we would try to align the samples, we see that we would need not only linear transformations, but also and non-linear warps. This hints that already the built-in 2D polynomial calibration routines of both eye-trackers differ in their estimated calibration coefficients, even though they are quite similar from an algorithmic point of view.

In contrast to the good performance of both eye-trackers in the accuracy task (section “Results Task 1/7/10: Accuracy Task with Small Grid I and II”), we see qualitative differences in the Free Viewing analysis. Especially the bad fusion of the eye positions and the high variability of the samples recorded with the Pupil Labs glasses are obvious. In addition, Pupil Labs finds fewer and shorter saccades than EyeLink and therefore on average longer fixation durations. Hence, the eye-tracker should be carefully chosen, if individual eye traces are of importance.

Results task 4: microsaccades

If saccade-like behavior is found while the participants subjectively fixates, they are usually termed microsaccades. In order to investigate how well microsaccades can be found, we showed a central bullseye fixation point for 20 s to elicit these microsaccades and analyzed their amplitudes and rates (Fig. 10).

Figure 10 (A) Microsaccades task.

(B) The participants kept fixating a central fixation point for 20 s. (C) Microsaccade amplitudes. (D) Microsaccade rates. (E) Main sequences for both eye-trackers. Different colors depict different participants.

For EyeLink, the winsorized mean amplitude was 0.23° (IQR: 0.18–0.28°), for Pupil Labs 0.18° (IQR: 0.15–0.23°), with a paired difference of 0.03° (CI95 [0.08 to −0.02°). These microsaccade amplitudes follow what is expected from pupil-estimated microsaccades (Nyström et al., 2016). The microsaccade rate is also in line with previous research (Winterson & Collewun, 1976; Rolfs, 2009). For EyeLink, the winsorized mean number of microsaccades was 117.2 (IQR: 79.5–165.5), for Pupil Labs 66.73 (IQR: 35.0–98.0), with a paired difference of 47.0 (CI95 [75.67–16.20]). This indicates that Pupil Lab finds only ≈50% of microsaccades.

The main sequence of the Pupil Labs glasses shows much higher variance (Fig. 10E), while the main sequence is cleanly visible in the EyeLink plot. Unsurprisingly, it is difficult to identify microsaccades in the Pupil Labs data. Even though the amplitudes of reported microsaccades look comparable, the number of microsaccades is much reduced.

We made use of the co-measurement of both eye-trackers and tried to pair the classified microsaccades. Using the Eyelink 1000 we found over all participants and blocks 1,788 microsaccades, using Pupil Labs glasses only 1,105. Approximately 600 of those co-occur in the same time-window (overlapping time windows or time windows within each other). Taking the EyeLink 1000 as the reference would imply that using the Pupil Labs glasses we detected only approximately 33% (6001,788) of all possible microsaccades and we would observe a high number of false positives (1,105−6001,105≈45%). Taking the jointly detected microsaccades as the reference would imply a high rate of misses and false alarms for both systems. The truth is probably in-between, but a definite answer cannot be given due to the lack of ground truth.

It is very reasonable to assume that the analysis of the EyeLink data will also have many false positives, but likely less than the with the Pupil Labs glasses, given the better precision value of the EyeLink 1000. We can only conclude that detecting microsaccades is challenging for both eye-trackers, but the Pupil Labs glasses are most likely not up to the task.

In the grid task, Pupil Labs often missed small corrective saccades (Fig. 7). In the Free-Viewing task, we observed longer fixation durations for the Pupil Labs glasses which readily can be explained by missed small saccade amplitudes as well. Therefore, it is unsurprising that Pupil Labs also has problems with classifying microsaccades, and in addition, similarly to the Free Viewing task, reports them as shorter as our reference eye-tracker.

Results task 5: blink task

In this task, we asked participants to voluntarily blink after a short beep (Fig. 11).

Figure 11 (A) Blink task.

(B) The participants blinked after they heard a beep sound which was repeated seven times. (C) Blink durations. The eye-trackers’ built-in blink classification algorithms were used. Each individual blink is shown in green. (D) Number of reported blinks.

For EyeLink, the winsorized mean number of blinks was 7.1 (IQR: 7.0–7.33), for Pupil Labs 5.3 (IQR: 3.9–6.7), with a paired difference of 1.8 (CI95 [3.1–0.8]).

For EyeLink, the winsorized mean duration of a blink was 0.190 s (IQR: 0.154–0.240 s), for Pupil Labs 0.214 s (IQR: 0.170–0.257 s), with a paired difference of −0.025 s (CI95 [−0.004 to −0.039 s).

Typical voluntary blink duration is found to vary from 0.1 s to 0.4 s, with longer blinks reported from Electrooculography electrodes than by eye-trackers (VanderWerf et al., 2003; Benedetto et al., 2011; Riggs, Volkmann & Moore, 1981; Lawson, 1948). In the EyeLink data, we seems to recover all seven blinks and some additional (likely spontaneous) ones. In contrast, Pupil Labs current blink classification algorithm is not sufficient to reliably classify eye blinks. We even had to modify their blink classification algorithm (see section “Eye Movement Definition and Classification”) in order to use it in the first place. Nevertheless, all seven blinks were detected correctly for some participants also with the Pupil Labs glasses, but not on the group level.

Results task 6: pupil dilation task

We used four luminance levels to constrict the pupil, each which was preceded by a black baseline stimulus (Fig. 12).

Figure 12 (A) Pupil Dilation task.

(B) We showed participants four different luminance levels for 3 s each. Prior to each luminance, we showed a black baseline for 7 s. (C) Change in normalized pupil area relative to median baseline for the four different luminance levels for both eye-trackers (left and right facing triangles). Data were time-binned prior to plotting. Winsorized mean over participants of the winsorized means over blocks with 95% bootstrapped confidence intervals for each eye-tracker. (D) Winsorized means and 95% bootstrapped confidence intervals of the pupil area for each luminance level, average of 2–3 s after luminance change. Each left summary statistic of a pair depicts the Pupil Labs glasses, the right the Eyelink 1000. (E) Difference in normalized pupil area between the eye-trackers. Each blue line refers to the winsorized mean of one luminance level of one participant. The aggregated data over subjects (gray line) illustrates that the measurements of the eye-trackers differ little on an aggregated level, but subject-wise the eye-trackers do estimate the size of the pupil area very differently.

On the group level, both eye-trackers seem to measure the same normalized pupil area (see Figs. 12C and 12D). However, looking at the estimates of pupil area per participant (Fig. 12E), we observe that each of the eye-trackers has a reliable subject-specific bias. Due to this discrepancy between single subject and group level pupil dilation, researchers should be careful when relying on individual participants’ pupil dilation. However, on the group level, we think that there will be not much difference in using either eye-tracker.

Results task 8/task 9: head movements

Results yaw movement

Eye movements rarely occur without head movements. Therefore, we let participants move their head with their nose (and centered gaze) pointing to fixation targets presented on a horizontal line. In total we used five different target positions (Fig. 13).

Figure 13 (A) Head yaw task.

(B) The participants rotated their head so that their nose pointed to one of five horizontal targets. The participants pressed the space bar after they finished the head movement. (C) Single subject plots: distance of mean fixation to target fixation. An ideal fixation would cluster around (0, 0). Constant offsets over all rotations, as well as systematic dependencies on the rotation angle can be found. Luminance indicates the position on the monitor (left: dark, right: bright). (D) Deviation in horizontal gaze component (E) and vertical gaze component of the estimated gaze position to the target position (red). For comparison, results of small grid I & II are also included. The plots show the winsorized means over participants and blocks with a 95% confidence interval. Light points show the winsorized mean over the blocks for a single participant.

We observed that in the EyeLink data, the horizontal component of the gaze relatively was estimated accurately, but the vertical component showed a systematic bias (compare Figs. 13D and 13E). The individual traces (Fig. 13C) show that half of the EyeLink participants have this pattern. Other participants are diffuse, either showing no effect of yaw movement or other idiosyncratic effects.

In contrast, Pupil Labs’ patterns are different. Here, we find a systematic, larger effect in the horizontal component. In addition, the vertical gaze component shows a positive offset for all target positions. It could be possible that physical slippage of the glasses during the task or experiment due to head motion could be the reason for this offset in vertical accuracy. Both systematic biases can be found in reduced strength in the small grid conditions. Interestingly, the two small grid conditions, before and after the two head movement blocks, seem to be indistinguishable. This is a hint that the systematic effect we see during the yaw-task is a dependency on head position and not pure slippage.

Results roll movement

Similar to the head yaw, we also investigated head roll. We instructed the participants to roll their head until their eyes were aligned with a line that we presented at angles ranging from −15 to 15°. During the roll movement, the participants were instructed to keep their fixation on a central fixation target (Fig. 14).

Figure 14 (A) Head roll task.

(B) The participants tilted their head until their eyes were aligned with a skewed line and kept fixating a central fixation target. The skewed line was presented at seven different angles from −15 to 15°. (C) Individual participants’ results. Mean fixation location is shown. In the ideal case, the points would be clustered around (0, 0). Luminance indicates the position on the monitor (counterclockwise: dark, clockwise: bright). (D) Deviation in horizontal gaze component or (E) vertical gaze component of the winsorized mean gaze position for all participants.

The EyeLink data showed a linear dependency of horizontal fixation position and head roll angle. The slope of this dependency differed between subjects from a negative slope to a slightly positive one. Interpreting the individual subject traces (Fig. 14C), it is clear that the vertical deviation is stronger in most participants. There seems to be no relation between the strengths of horizontal and vertical offset. For Pupil Labs, all slopes seem to be straight and we found only constant offsets. Conversely, in the individual participant traces Pupil Labs mostly shows a clustered but biased shape.

Both eye-trackers seem to have their own systematic problems with head movements. For traditional stationary experiments these problems can be ignored, for mobile setups with free head movements, these problems become much more important (cf. Niehorster et al., 2018). Taken together, we observed head movement biases of on average 1° for yaw and roll, with up to 3° in individual subjects. The resulting biased accuracy values deviate to a great extend from the typical accuracy values that we observed in the grid task.

Discussion

Summary

In this paper, we recorded participants’ gaze data using the EyeLink 1000 and the Pupil Labs glasses simultaneously in a newly developed eye-tracking test battery. The gaze data was used to analyze a multitude of eye-tracking related measures to compare the eye-trackers. Our test battery shows superior accuracy as well as precision values for the EyeLink 1000 compared to the Pupil Labs glasses (average accuracy: 0.57 vs. 0.82°; average precision (SD): 0.19 vs. 0.31°). Similarly, we measured the decay of calibration and the EyeLink 1000 was almost robust to this, while the Pupil Labs glasses showed a decay by 30% after 4 min 30 s. Having a variety of eye-tracking tasks in our test battery, we also looked at less typical performance measures. Our Free Viewing tasks allowed for more qualitative comparison and indeed, we found large differences between the signals: Visual inspection showed high variance of samples of the Pupil Labs glasses and quantitatively we found fewer and shorter saccades in the Pupil Labs glasses data and therefore also fewer fixations than with the EyeLink. The effect of smaller amplitudes is also reflected in other measures, for example, a smaller rate of reported microsaccades. Our test battery allows us to also look at the performance of blink classification and here we found accurate eye blink classification by the Eyelink 1000 but not the Pupil Labs glasses. Looking at the impact of head movement on the recorded gaze signals, we found that both eye-trackers were equally susceptible to head motion: the EyeLink 1000 is more vulnerable to roll movements and the Pupil Labs glasses more to yaw movements. We also observed that with both eye-trackers, pupil dilation seems to be recorded equally well on the population level, but subject-wise, robust eye-tracker differences exist. Likewise, we did not find large group differences between the eye-trackers in our model based task-specific smooth pursuit analysis. This set of differences and similarities shows the importance of a heterogeneous test battery to compare eye-trackers.

Accuracy

Accuracy is the dominant metric to evaluate eye-trackers, but as a single metric, it cannot summarize performance for all typical eye-tracking experiments. Nevertheless, it is very useful and correlates with many other evaluation metrics. We first discuss the results of the EyeLink 1000 followed by the Pupil Labs glasses.

Our measured winsorized mean accuracy for the EyeLink 1000 was 0.57° (which is larger than the manufacturer-specified accuracy of <0.5°). Comparing our measured value of the EyeLink 1000 accuracy to values reported in the literature, we found comparable values from Barsingerhorn (2018) who found mean accuracies of 0.56° horizontally and 0.73° vertically for the EyeLink 1000. However, we also also encountered much worse accuracy values from Holmqvist (2017) who reports an accuracy of ≈0.97° for the EyeLink 1000 in a study comparing 12 eye-trackers, but they did not select for ideal conditions. Our measured accuracy for the Pupil Labs glasses (0.82°) is larger than the manufacturer specified accuracy of 0.6° (N = 8, Kassner, Patera & Bulling, 2014) and similar to a recent study comparing wearable mobile eye-trackers (N = 3, MacInnes et al., 2018) who reported 0.84°.

Given these accuracy results, researchers now can take consequences for their own studies. For instance, in a region of interest (ROI) analysis, they can make sure that their ROIs are much larger than the eye-trackers fixation spread and accuracy. Orquin & Holmqvist (2018) offer such a simulation to test how large the ROI needs to be, dependent on the precision of the eye-tracker. During the design of one’s own studies, one should perform these simulations and see for themselves if the paradigm, the size of ROI, or the device has to be changed.

Often researchers use a manufacturer calibration-validation procedure to get an estimate of the accuracy. To validate such a procedure, we can compare the manufacturer values to our own results (which were measured immediately after the manufacturer ones’): The EyeLink 1000 manufacturer validation procedure accuracies were better than our own accuracy estimates (0.35 vs. 0.57°)3 . At first sight, this is surprising as the EyeLink 1000 software uses a similar procedure to our grid task (compare section “Task 1/Task 7/Task 10: Accuracy Task with the Large and the Small Grid”) for their calibration/validation procedure (according to the SR-support). In the EyeLink 1000, saccades are first detected in order to find a stable fixation and calculate the mean fixation position, then the Euclidean distance to the validation target is calculated. This is analog to our analyses, except that we make use of the spherical angle instead of the Euclidean distance on the screen. We also do not make use of slight changes of distance to the monitor due to subject motion (which the EyeLink 1000 does by measuring the size of the head-marker). Yet another difference is, that in the EyeLink 1000, average accuracy is calculated as a weighted average (SR-Support forum), weighting the central point (with generally best accuracy) with the same amount as the sum of all four corner points (with generally worst accuracy)4 . When we restricted our analysis to the same test locations and used the same weighted average we did indeed find better accuracy, but only slightly so (0.54–0.57°) and still far off from the EyeLink reported accuracy of 0.35°. Unfortunately, data from the manufacturer validation procedure cannot be recorded simultaneously. Consequently, we currently do not know how the deviation in accuracy values arises.

Interestingly, Pupil Labs’ own validation procedure reported worse accuracies (1.04°) than what we subsequently measured. In their case, this might be the result of their differing accuracy calculation routine. Instead of selecting one fixation, they use every sample reported while the validation target is visible. They then exclude samples too far from the target and the offset between the average of all remaining samples to the displayed validation point are used to estimate the accuracy value. Hence, this calculation results in a very conservative estimate as most likely some samples during the saccade or from undershoot fixations are still included.

In summary, we found accuracy values that are worse than the manufacturer advertised ones, but overall, the accuracy values were in a very good range for eye-tracking research.

Results in the light of common experimental paradigms

Our main motivation for this study was to offer many different measures of eye-tracker performance to be able to evaluate the requirements of many individual experimental paradigms. In a simple two-images choice paradigm, both tested eye-trackers seem equally suited to measure first fixation location and saccadic reaction time (Cludius et al., 2017) if the images are large enough (usually such images are at least 5°). Switching to more natural tasks like free viewing, one can see big differences between the eye-tracker in the quality of the signal of the individual traces. While the aggregation in the Grid task shows good performance, visual inspection of the Free Viewing task tells a different story. The Pupil Labs glasses exhibit much higher variance especially visible during saccades. This makes the interpretation of single traces on free viewing paradigms difficult, but aggregated measures (e.g., salience maps) could still be interpretable (Waechter et al., 2014).

Smooth pursuit eye movements are very common when moving through the world or watching movies (or other dynamic stimuli). Our test battery tests smooth pursuit in a formal way and in this paper we analyze smooth pursuit using a formal model as well. This is due to the current lack of applicable smooth pursuit classification algorithms that are compatible with our data (but see recent exceptions; Larsson et al., 2015; Pekkanen & Lappi, 2017; Bellet et al., 2018). We think that the smooth pursuit findings should be treated with caution as our analysis might not generalize to more natural conditions and unstructured smooth pursuit detection algorithms. Assuming they would indeed generalize, then both eye-trackers seem to be able to classify smooth pursuit reliably.

If blink classification is important in an experiment, for example, as a proxy for dopamine-related cognitive functions (Riggs, Volkmann & Moore, 1981; but see Sescousse et al., 2018), then Pupil Labs should not be used, or a new or custom blink classification algorithm has to be developed to report blinks reliably.

Other experimental paradigms have even higher requirements: one class of examples are EEG/eye-tracking combined studies which usually need very high temporal resolution to calculate fixation locked signal averages (Dimigen et al., 2011; Ehinger, König & Ossandón, 2015), where even the small differences in fixation onsets, which we found for Pupil Labs (see Fig. 7K), will result in a significant signal to noise ratio reduction.

We were initially positively surprised on the number of classified microsaccades in the Pupil Labs glasses’ data. But quantitative analyses showed that only around 55% of reference microsaccades were found. This assumes that the EyeLink 1000 reference eye-tracker can capture microsaccades adequately (but see Poletti & Rucci, 2016; Nyström et al., 2016). Taken together with the qualitatively noisy main sequence it seems unlikely that more microsaccades can be recovered by decreasing the microsaccade detection threshold of the algorithms or filtering the signal. It might simply be, that the spatial precision of the Pupil Labs glasses is not high enough for microsaccade studies.

For pupil dilation studies (Mathôt, 2018; Wahn et al., 2016) the eye-trackers do not seem to differ on the group level. We investigated maximally large effects (black to white) and found reliable differences for pupil dilation between the eye-trackers only on the individual subject’s level. But as most experiments are interested in the group-level, this finding should not pose a problem.

Head yaw, a very common head movement, posed a problem for both eye-trackers. The consequences were not extreme, but notable (≈1° additional error for a large rotation of 40°). Head roll had a systematic effect only on the remote EyeLink 1000 but not the Pupil Labs glasses. In a related study, Niehorster et al. (2018) also investigated yaw and roll in a diverse set of remote eye-trackers. In contrast to our study, they used the most extreme head movement while still being able to fixate a dot. In accordance, their resulting effects of yaw and roll are much stronger for some of their participants than what we observed.

These interpretations are of course not exhaustive but show how such a diverse test battery allows to evaluate eye-trackers on a task-individual basis. This study allows researchers to plan and test their eye-trackers using our test battery, and once more eye-trackers have been tested, select the eye-tracking equipment according to the design of their studies.

Mobile settings

As mentioned above, all of our results are based on data which were recorded under optimal lab conditions (in contrast to a mobile setting where the subject is freely moving. Therefore, we offer a lower bound for accuracy and only a rough basis for extrapolation to more mobile setups. In realistic mobile setups, the calibration decay we observed will likely be worse as head movements (and therefore slippage) increase. It is also possible that the 3D-eye algorithm offered by Pupil Labs provides higher stability over time at the cost of overall worse accuracy, as it is advertised as no slippage albeit on the cost of accuracy. This needs further testing. In general, there are many reasons that make the analysis of mobile recordings more difficult: firstly, the parallax error which occurs if one uses a scene camera and fixations change in depth (Mardanbegi & Hansen, 2012; Narcizo & Hansen, 2015). Secondly, uncontrollable luminance differences, which directly influence pupil dilation and bias the estimated gaze position (Brisson et al., 2013; Drewes et al., 2014). Thirdly, head movements, which we showed also in this experiment (Cesqui et al., 2013) and fourthly, due to large saccades to eccentricities outside of the calibration range. This is by no means an exhaustive list, but just four reasons as to why one will encounter difficulties when going into mobile settings. Further comparisons in mobile settings and with mobile eye-trackers are needed (see a recent study with N = 3 comparing three wearable mobile devices MacInnes et al., 2018).

Eye-tracking test battery

Our eye-tracking test battery proved to be useful and comprehensive in this eye-tracking comparison study. In case anyone would like to use the test battery to evaluate other eye-trackers, we recommend several small changes in experimental design and analysis:The smooth pursuit analysis should be based on an analysis method that classifies smooth pursuit without the prior information of smooth pursuit direction (Larsson et al., 2015; Pekkanen & Lappi, 2017; Bellet et al., 2018). We tried to detect smooth pursuit directly and implemented the NSLR HMM algorithm (Pekkanen & Lappi, 2017) into our pipeline. But the results were not usable for the Pupil Labs glasses, whereas the EyeLink 1000 was doing better (the bad fusion of eyes could be one explanation, but we did not investigate further). We also found a very high number of catch-up saccades even though following procedures described by a previous study. This needs further investigation.

Some eye movement behaviors are missing from the test battery: for example, vergence (but see Hooge, Hessels & Nyström, 2019) calibration/validation in depth and nystagmus. Especially for mobile setups (or VR-environments) calibration in depth would be very interesting to evaluate.

Two changes should be made to the pupil task: first, a follow-up study should try to measure the true pupil size in addition to the one reported by the eye-trackers. With this the individual subject differences we observed could be studied in greater detail. Second, the influence of pupil dilation on the gaze signal could be tested by repeating the procedure at multiple fixation locations (Brisson et al., 2013; Drewes et al., 2014).

In conclusion, it is clear that our eye-tracking test battery offers an extensive description of most eye movement parameters and other missing parameters can easily be included in future versions.

Pupil Labs: ongoing development and challenges

The software and algorithms employed by Pupil Labs are continuously developed and improved. This means that this comparison paper will always be outpaced by the new methodologies offered by Pupil Labs and we can only test a snapshot of development. We want to point out that Pupil Labs offers the full raw eye-videos, and any old analysis can, in principle, be updated with newer algorithms and software. Our own analysis pipeline makes use of Pupil Labs’ code and can be updated on demand. This is slightly complicated by Pupil Labs, as they do not offer an official API, but one has to access the code of the GUI-based software. Therefore, no guarantees for software compatibility over versions exist and our pipeline (and those of other researchers) could break once Pupil Labs updates their algorithms. Therefore, we recommend sticking to one recording and one analysis software version for the whole project. We also want to note that the current GUI-based analysis software is easy for consumer use but quite difficult to use for reproducible research. Using the GUI, many manual steps are necessary for each participant, to go from eye-video recordings to accuracy values. Our own pipeline makes use of Pupil Labs’ open source code and circumvents these problems. To facilitate research with Pupil Labs, we offer our own makefile to automatically compile most dependencies to run Pupil Labs from source, without the need for root-rights.

We noticed two problems with Pupil Labs’ algorithms that could be directly improved upon. First, blink classification was a problem, as the Pupil Labs algorithm relies on the change in pupil-confidence (section “Eye Movement Definition and Classification”) instead of an absolute signal (e.g., EyeLink uses a fixed number of frames without pupil detection). We had to improve their algorithms, since we were often loosing large chunks of data (10’s of seconds) to the failing blink classification algorithm. Second, poor fusion of binocular eye gaze streams. We recorded binocularly, but often it seems that the reported trajectory show eye-individual calibrations rather than binocular fusion (see section “Discussion” and Figs. 9C and 9D). Thus, a high variance orthogonal to the saccade trajectory is introduced. The poor fusion of the eye gaze streams is also reflected in the high standard deviation precision value. On one hand, this problem is likely influencing velocity-based saccade classification algorithms like the one we used in this study. On the other hand, it is unlikely that this problem influences the accuracy estimate, as we use fixation-wise mean gaze positions. Another phenomenon related to bad fusion can be observed more in the temporal domain: while the reported sampling frequency is 240 Hz, in practice, the effective sampling rate ranges from 120 to 240 Hz (see Fig. 4). It is possible that future revisions of the software will fix these problems. In the present study, the Pupil Labs eye-tracker served as a comparison to our reference. As to be expected in such a comparison, both accuracy was worse and spatial precision smaller. Small saccades were sometimes, blinks often, missed. But the average accuracy was well below 1° visual angle and pupil dilation could be resolved as good as with our top-of-the-line reference (as far as we can tell from our data). Thus, taken together, it appears that the Pupil Labs eye-tracker is a valid choice when mid-range accuracy is sufficient, lab conditions are present, repeated calibration is possible, medium-to-long saccades are to be expected and one does not rely on the accurate classification of blinks.

Limitations of the present study

Our comparison study is limited, especially in how well it extrapolates to other situations. We used only healthy, young, educated, western participants with 6/6 vision. And even from those we only included ≈70% in the study and rejected the others, as we could not calibrate them with both eye-trackers concurrently. In a more diverse population, there are participant groups whose eye movements are notoriously difficult to measure, for example children, elderly participants or some patients suffering from autism (compare Barsingerhorn, Boonstra & Goossens, 2018). The performance when measuring a less homogeneous population remains to be measured, but will likely be worse than our sample here. Therefore, we want to stress again that our study reproduces a typical lab setup. In more advanced setups, for example, mobile or VR studies, the performance will also be likely worse due to more head movements.

The choice of the eye movement event classification algorithm could have large influence on most of our results. In this study, we used a very popular velocity based saccade classification algorithm (Engbert & Mergenthaler, 2006). This algorithm was developed for eye-trackers from SMI and SR-Research as one of the most accurate video-based eye-trackers for lab setting research (Holmqvist, 2017). It is therefore possible that there is a bias against Pupil Labs when using this algorithm. Pupil Labs offers their own fixation classificator based on spatial dispersion, but informally we found it lacking in many situations. The classification algorithm could have large effects on some of our findings, for example, precision, smooth pursuit speed, fixation number, and duration. However, we think that the effect on spatial accuracy will be small. Indeed, using a new algorithm based on segmented linear regression and a hidden markov model (Pekkanen & Lappi, 2017), we found near identical spatial accuracy results, but the results for the precision measure and several others (e.g., the number and duration of the fixations in the Free Viewing task) changed a lot. Future comparison between algorithms and eye-trackers could also make use of more sophisticated event-matching algorithm (Zemblys, Niehorster & Holmqvist, 2018), differentiating, for example, between split and missed events. In general, the comparison of algorithms is not the focus of this article and has been done in other studies (Andersson et al., 2017).

There are more factors that might have given us non-optimal measured performances in our study: The experimenter recording the data had less than a year of eye-tracking experience; we had to calibrate two eye-trackers at the same time; and, at least for the EyeLink 1000, the calibration area on the monitor was slightly larger than what is recommended (we used 36° with a recommended range of 32°). We argue that these points cannot be critical, as we easily reached the manufacturer recommended validation results. In addition, throughout the study we used robust statistics to mitigate the influence of singular outliers.

All in all, we think none of the limitations are so critical as to invalidate our findings.

Conclusion

Eye-tracking data quality cannot be reduced to a single value. Therefore, we developed a new test battery that allows to analyze a variety of eye-tracking measures. We used this test battery to evaluate two popular eye-trackers and compare their performance. We exemplarily interpreted our findings in light of many popular eye-tracking tasks and thereby offer guidance on how to interpret such results individually for the researchers own tasks/eye-tracker combination.

Supplemental Information

Supplemental Information 1 Overview of Participant Information and Rejection.

Click here for additional data file.

We thank Daniel Backhaus and Hans Trukenbrod for discussions, experimental code and stimulus material for the free viewing task. We thank Anna and Ida Gert for valuable input throughout the project.

Additional Information and Declarations

Competing Interests

Author Contributions

Human Ethics

Data Availability

1 The instruction texts can be found on GitHub https://github.com/behinger/etcomp/tree/master/experiment/Instructions.

2 In case you noticed a mismatch between the paired difference of winsorized means and the winsorized paired difference: meanw(X) − meanw(Y) ≠ meanw(X − Y) because different values are being winsorized.

3 After publishing a preprint of this manuscript, SR Research reached out to us to discuss this discrepancy, their full comment can be found on https://www.biorxiv.org/content/10.1101/536243v1.

4 Four corners with weight 1, four edges with weight 2, four middle points with weight 4 and the central point with weight 10.

Peter König is the chief scientist of WhiteMatter Labs, an eye-tracking related company. The Institute of Cognitive Science owns seven Pupil Labs eye-trackers and one EyeLink 1000. We are an independent research group with no ties to the Pupil Labs company or to SR Research. The manufacturers made no recommendations to the design or analysis of the study.

Benedikt V. Ehinger conceived and designed the experiments, performed the experiments, analyzed the data, contributed reagents/materials/analysis tools, prepared figures and/or tables, authored or reviewed drafts of the paper, approved the final draft.

Katharina Groß analyzed the data, contributed reagents/materials/analysis tools, prepared figures and/or tables, authored or reviewed drafts of the paper, approved the final draft.

Inga Ibs performed the experiments, wrote the experimental software, approved the final draft.

Peter König conceived and designed the experiments, authored or reviewed drafts of the paper, approved the final draft.

The following information was supplied relating to ethical approvals (i.e., approving body and any reference numbers):

The ethics comittee of Osnabrück University granted Ethical approval to carry out the study (4/71043.5).

The following information was supplied regarding data availability:

Data is available at Figshare:

Ehinger, Benedikt (2019): EyeLink 1000 data and Pupil Labs Videos (∼700 GB). figshare. Collection. DOI 10.6084/m9.figshare.c.4379810.v2.

Code is available at Zenodo: DOI 10.5281/zenodo.2553447.

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
