# Peer review of "A new comprehensive eye-tracking test battery concurrently evaluating the Pupil Labs glasses and the EyeLink 1000"

_PeerJ, doi:10.7717/peerj.7086_

## Round 0.1 · original submission · Major Revisions

Both reviewers see the relevance of your manuscript and the proposed battery of tests, but they also raised a substantial amount of critical points, which must be addressed before publication. As one of the reviewers notes, the revised manuscript should increase focus of the presentation and reduce redundancy. Discussion of results should be sharpened as well.

Reviewer 1 ·

Basic reporting

See general comments

Experimental design

See general comments

Validity of the findings

See general comments

Additional comments

In this paper, Ehinger et al. introduce a test battery in order to compare different eye trackers with one another. They then demonstrate the applicability of this battery in a comparison between the EyeLink 1000 and the Pupil Labs Glasses. Overall I am positive regarding this manuscript. There aren’t many eye-tracker tests in the literature, and any efforts to improve this are welcome. However, I also have some critical comments, that I believe should be addressed in order to make this manuscript publishable. While there are many technical details throughout the manuscript (which I generally applaud), it is often unclear what problem these techniques solve (which should be stated up front). In other cases, there is information that doesn’t add anything, and quite a few repetitions occur. Addressing these issues may shorten the methods/results section, which is quite lengthy at present. Second, I find the framing of this test battery lacking in terms of its motivation and its comparison to previous work. I outline all my comments below. Given that it is a long paper, and I have many comments, I will note them as they first appear in the paper. As such, the comments are not ordered by importance. I will explicitly mention when points are essential to me.

Introduction p.1 line 44: “a single index will not be sufficient to characterize the suitability of an eye tracker for a specific application, but a more comprehensive test is needed.” I disagree. A single index can be sufficient, it just depends on the application. And each application may require a different measure to address suitability. Please nuance: a comprehensive test can inform suitability for more applications.

Introduction p.2. The paragraph from line 48-66 is unnecessary. Instead, I urge the authors to explain the different metrics they investigate, and highlight with example from the eye-tracking literature where they are important. For example, high accuracy may be important for reading research, high precision for microsaccade research, robustness to head movement for infant research, etc.

ESSENTIAL: No clear definitions of fixations, saccades, smooth pursuit, etc. are given. Upon introducing the different eye-movement terms such as fixation, saccade or smooth pursuit, the authors should refer to:
Hessels, R. S., Niehorster, D. C., Nyström, M., Andersson, R., & Hooge, I. T. C. (2018). Is the eye-movement field confused about fixations and saccades? A survey among 124 researchers. Royal Society Open Science, 5(180502), 1–23.
In this paper, semantic disagreement among eye-movement researchers is characterized and the different ways of defining eye movements are discussed. I believe it is essential that the authors give clear definitions as well, as it directly impacts on their work. For example, in the Introduction p.2, lines 67-70, the authors write “It is clear that such a range of paradigms … that types in a controlled manner.”
I agree with the second sentence in this short paragraph. However, I disagree with the first. What is considered by researchers as a specific “eye movement” may differ depending on the research setting as is shown by Hessels et al. (2018).
As a second example, a few lines later (73), the authors write “even when participants think they fixate …”. Here what is considered a fixation is of vital importance. There are many examples in the literature of where fixation is considered to happen when participants follow the instruction to fixate, as such drift etc. are part of the fixation as “fixational eye movements”. This should be clarified.

Introduction p.2 lines 86-88. These sentences are unclear. In my experience, the pupil labs (~1000 euro) and the EyeLink 1000 (~30.000+ euro) can vary much more than a factor 15, or is that not what is meant? I also don’t which “features” the authors refer to in the last sentence.

IMPORTANT: Introduction p.2 lines 89-90. Please acknowledge the work that IS available on the performance of eye trackers from independent researchers. These are, at least:
Blignaut, P., & Wium, D. (2013). Eye-tracking data quality as affected by ethnicity and experimental design. Behavior Research Methods, 46(1), 67–80. http://doi.org/10.3758/s13428-013-0343-0

Hessels, R. S., Cornelissen, T. H. W., Kemner, C., & Hooge, I. T. C. (2015). Qualitative tests of remote eyetracker recovery and performance during head rotation. Behavior Research Methods, 47(3), 848–859. http://doi.org/10.3758/s13428-014-0507-6

Niehorster, D. C., Cornelissen, T. H. W., Holmqvist, K., Hooge, I. T. C., & Hessels, R. S. (2018). What to expect from your remote eye-tracker when participants are unrestrained. Behavior Research Methods, 50(1), 213–227. http://doi.org/10.3758/s13428-017-0863-0
The latter two also make the explicit point that accuracy and precision values are not sufficient to characterize eye trackers for e.g. infant research in their case and compare robustness to movement of many eye trackers. Niehorster et al., 2018 also test the EyeLink 1000 in explicitly during yaw and roll movements, and the findings of the manuscript should therefore be directly compared to the findings in Niehorster at al.

Introduction p.2 line 69. “Calibration decay” and Introduction p.2. line 97. “Decay of accuracy”.
Please define decay of accuracy here. There is also empirical work on the decrease of accuracy over time that is missing here:
Nyström, M., Andersson, R., Holmqvist, K., & van de Weijer, J. (2013). The influence of calibration method and eye physiology on eyetracking data quality. Behavior Research Methods, 45(1), 272–288. http://doi.org/10.3758/s13428-012-0247-4

Methods p.3 line 114. “Section 1”, should be supplementary materials I presume?

Methods p.4 line 143. Round trip delays are noted to be <1 ms. I assume that the authors do not claim that a gaze coordinate is calculated from the video images within the system below 1 ms? Please clarify what this sentence adds, and what problem is solved.

Methods p.4 lined 152-153: change author names to initials, write out < and > signs, e.g. “with more than 5 years experience”

Methods p.4 line 162 “in their own time” -> “at their own pace”

ESSENTIAL: Methods p.4 line 164. “each task either measures attributes of the eye-tracking devices (e.g. accuracy) …”.
This is wrong. Accuracy is an interaction between at least the eye tracker, the participant and the operator, and the environment (see e.g. Nyström et al., 2013 mentioned above).

Methods p.4 line 168-169. Sentence on task sequence is redundant, please remove.

Methods p.5 line 180-181. “the eye movement classification algorithms for event detection”. How do these differ? See also Hessels et al., 2018 for a discussion on the problems inherent in the term “detection”, and the preference for using “classification”. Please reconsider all mentions of “classification” and “detection”.

Methods p.5 line 185. “Maximal manufacturer filter settings”. Make this explicit. What filters and what settings?

IMPORTANT: Methods p.6 lines 195-200. The gaze coordinates of the Pupil Labs are mapped to a world-centered coordinate system. In combination with the fact that the participant does not move around, this means that you consider the Pupil Labs as a remote-like eye tracker here. Please make this explicit (saying that it is a lab-setup is not explicit enough). In this manner, it doesn’t differ much from e.g. the EyeLink 2 head-mounted eye-tracker, which also reports gaze in screen coordinates. Nor does it mean that your findings will generalize to situations including locomotion.

Methods p.6 section Blink detection. From this section it seems that the authors are not only comparing eye trackers, but also blink detection methods. Why do the authors not conduct the blink detection for the EyeLink data themselves as well?

Methods p.7-8 lines 254-279. This section is unclear. Moreover, it is unclear what problems are being solved. Please revise: start with the problem, why it is important for the measures, and how it is solved. Optionally, this could be moved to an appendix.

Methods p.8 line 282. “Actual target point”, or “Instructed target …”

Methods p.8 line 293. Please add a link to the reference for Holmqvist 2017. It is not included in the reference list.

Methods p.8 line 297. Spatial precision is often noted as the variable error in the gaze coordinate signal. This seems a better description than “consistency of samples”. What is meant by consistency?

Methods p.9 line 306. “Task 2.3.9” doesn’t exist, it is a section.

Methods p.9 line 318. “The task sequence…”. This is the third iteration, please remove.

Methods p.9 line 337. “Fixation were manually accepted”. Calibration points were manually accepted I assume? Fixations doesn’t seem like the right word, unless the Pupil Labs software returns a set of “fixations” from the eye-tracking data that the user can choose from.

IMPORTANT: In many cases, dashes (-) are missing in the writing. This is not grammatically correct, and impedes the reading of the manuscript. E.g. “13 point verification” -> “13-point verification” or “validation accuracy limit” -> “validation-accuracy limit”. This occurs in many locations. Please revise the manuscript critically for such instances.

Methods p.10. I like the randomization process!

IMPORTANT: Methods p.10 line 393. “It [smooth pursuit] is especially common while we move relative to a fixated object”. This is in stark contrast with the views held by many researchers, who hold that an object needs to move with respect to the world and continuously looked at before it is considered smooth pursuit. Movement of oneself with respect to the world is not considered smooth pursuit by many researchers. This is worked out in Hessels et al., 2018. This again emphasizes the importance of giving definitions, or mentioning where researchers are divided on the definitions.

Methods p.10 line 394. “Smooth pursuit detection is still in its infancy”. Please cite the works that HAVE done this, at least:
Larsson, L., Nyström, M., Andersson, R., & Stridh, M. (2015). Detection of fixations and smooth pursuit movements in high-speed eye-tracking data. Biomedical Signal Processing and Control, 18, 145–152. http://doi.org/10.1016/j.bspc.2014.12.008

Methods p.11 section “Measures” starting from line 410. It is unclear why this method is used. Please first describe the problem, and then how it is solved.

Methods p.11 line 431. Remove “a thesis evaluating … EyeLink 1000”

Methods p.11 line 445-446, please cite the relevant study (it is mentioned before).

Methods p.11 line 448. “the Engbert algorithm” -> “the Engbert & Mergenthaler (2006) algorithm”.

Methods p.11 line 453-454. Is there a specific reason? If not, the sentence doesn’t add much.

Methods p.12 line 474 “8 different”, shouldn’t that be 7?

IMPORTANT: In the Results section, every subsection starts with a paragraph repeating the Methods section. This unnecessarily lengthens the manuscript. Please remove, shorten or revise.

Results p.16 line 531. What does the “(4)” refer to?

Results p.16 line 569. How can it be that the difference is -0.001s while the means are 0.004 apart? Maybe I missed something.

Results p.19 line 606. “Significant shortcomings”. Who/what is the standard here? Keeping the sentence neutral (i.e. not using “shortcoming”) is already informative enough.

Results p.19 line 611. “The eye never stays still”, please define the reference frame. With respect to what?

IMPORTANT: Results p.20 line 623. “Pupil Labs has problems identifying microsaccades”. This is wrong. The pupil labs doesn’t do this, the researchers do this using some algorithm. “It is not possible or problematic to classify microsaccades in pupil labs data” would be an accurate description. There are many such instances throughout the manuscript. Please revise such that the Pupil Labs and the EyeLink 1000 are not the acting agent. See e.g. also line 704.

Results p.21 line 643. “Blinks were detected correctly”. Please specify the benchmark here. Is that the EyeLink 1000? Or the authors?

Results p.21 lines 645-647 are unnecessary.

Results p.21 Figure 12. The legend for plot D is not complete. I do not understand what the two different lines refer to.

Results p.21 lines 654-657 are unclear.

Results p.21 line 658. “For this reason”, please explicitly mention the reason, as it is the start of a new paragraph.

Results p.22 line 662-663. “In order to investigate eye movements in more natural paradigms”. What is meant by natural? There are many lab-based studies that have to deal with participant movement, e.g. research with infants or certain patient groups, see Hessels et al., 2015, Niehorster et al., 2018 as mentioned above.

Results p.22 line 665, aren’t there 5 different target positions?

Results p.22 Figure 13. Define systematic drift here, and how it different from a constant offset.

Results p.23 line 688. What are “Biases”? How do they differ from constant offsets or systematic drifts. Please carefully define these, or revise the terminology. To me, it is an offset, not a bias.

Discussion p.24 line 717. “Accuracy is the dominant metric …”. Says who? and why?

Discussion throughout: please replace the Barsingerhorn thesis citation with the actual papers, e.g. at line 724 is should be:
Barsingerhorn, A. D., Boonstra, F. N., & Goossens, J. (2018). Development and validation of a high-speed stereoscopic eyetracker. Behavior Research Methods. http://doi.org/10.3758/s13428-018-1026-7

Discussion p.24 line 732-733. The following paper gives a much better simulation and provides code for running your own simulation for AOI size:
Orquin, J. L., & Holmqvist, K. (2017). Threats to the validity of eye-movement research in psychology, Behavior Research Methods. http://doi.org/10.3758/s13428-017-0998-z

IMPORTANT: Discussion p.25 line 757. “Our main motivation for this study was to…” This does not seem to match the content of the manuscript. The authors do not determine whether different experimental paradigms have different requirements, they assume this. That is what the development of their test battery is predicated on. Please rephrase this carefully, and make sure that the introduction and discussion match in their description of the goal of the manuscript: introducing the test battery and giving a proof of concept by comparing two eye trackers.

Discussion p.25 line 769. The word “indebted” in the sentence seems grammatically incorrect.

Discussion p.25 line 771, please cite these recent exception, e.g. Larsson et al., 2015.

Discussion p.25 line 771-772, this sentence is unclear. I do not see how the catch-up saccades are linked to the first part of the sentence. Please revise.

Discussion throughout. The authors talk about “mobile settings”. In many fields of eye-tracking research, an eye tracker is considered “mobile” if it can easily be moved around labs. Please define this carefully when first mentioning mobile settings, and explain that “mobile settings” refers to situations with wearable mobile eye trackers (glasses) where the participants moves with respect to the world.

Discussion p.25 line 793-796, relate these findings to Niehorster et al., 2018 for the EyeLink 1000.

IMPORTANT: Discussion p.26 line 798. “This study allows researchers to plan and select the eye-tracking equipment according to the design of their studies”. This is incorrect. Only if the choice is between an EyeLink 1000 and a Pupil Labs in “remote” mode would this be possible based on the present manuscript. This manuscript present researchers with a test battery in order to test their eye trackers on a range of measures. Please revise.

Discussion p.26 line 809. “Due to uncontrolled pupil dilation changes”. What is meant by this exactly?

Discussion p.26 line 821. As stated before, there are more smooth-pursuit classification algorithms than only the Pekkanen & Lappi algorithm.

Discussion p.26 line 828. The authors here describe the use of added depth calibration or recording. Please note that there is quite some discussion on whether this is reliably possible at all with video-based pupil-cr eye trackers, see e.g.:
Hooge, I. T. C., Hessels, R. S., & Nyström, M. (2019). Do pupil-based binocular video eye trackers reliably measure vergence? Vision Research, 156, 1–9. http://doi.org/10.1016/j.visres.2019.01.004

Discussion p.26 line 848-849 / p.27 line 852. Please write out the two problems. Not “Blink detection: …”. E.g. “First, blink detection was a problem …”

Discussion p.27 line 854. “poor fusion of the eyes”. By whom? By the participant or the eye tracker. This part can easily be misunderstood by readers, please revise.

Discussion p.27 line 855. I did not see any standard deviation precision measures, only RMS.

Discussion p.27 line 865. Please add that the Pupil Labs in “remote” mode is what is meant here. When participants walk around, it may be very different.

Discussion p.27 line 875. How do you know the performance will certainly be worse? Please nuance. “It may be worse” or “Is likely to be worse”.

Discussion p.27 line 878. “Determining the detection algorithms…” This sentence does not make sense to me.

Discussion p.27 line 880-881. The statement about SMI and EyeLink being the best eye trackers needs to be backed up by facts or stated as an opinion. In many situations (e.g. when head movement can occur), they are NOT the best, and this is based on facts. E.g. Hessels et al., 2015 / Niehorster et al., 2018.

Discussion p.27 line 884. Please clarify how the algorithm output affects precision. It is usually considered to be exactly the opposite. See e.g.
Holmqvist, K., Nyström, M., & Mulvey, F. (2012). Eye tracker data quality: What it is and how to measure it. Proceedings of the Symposium on Eye Tracking Research and Applications - ETRA '12, 45. http://doi.org/10.1145/2168556.2168563

·

Basic reporting

Important citations are missing.

Experimental design

Nothing to object against

Validity of the findings

No comment

Additional comments

Review of A new comprehensive Eye-Tracking Test Battery 1 concurrently evaluating the Pupil Labs Glasses and the EyeLink 1000 (Benedikt Ehinger, Katharina Groß, Inga Ibs, Peter König )

This is an interesting paper that has collected a number of paradigms into a test battery, and then evaluate the battery on co-recorded EyeLink and Pupil lab eye-trackers. I think the test battery is an interesting proposal, that gives this paper its major merit. I recommend publication, but would like to see a revision based on the comments I provide below.

Few studies have been made that co-record participants with two eye-trackers in order to test data quality. Surprisingly, this manuscript neither cites Jan Drewes co-measurements of the EyeLink with coils, or the co-recording comparison of the SMI RED with the EyeTribe by Agnes Scholz and colleagues. Co-recording with two VOGs is particularly tricky, as the infrared illumination of the two systems onto the same eye may interfere with the measurement of gaze in both of them.

p2, l76-78: I strongly disagree with the implied statement that the EyeLink is a gold standard eye-tracker. The only currently existing gold standards in eye-tracking are scleral search coils and the DPI. See Collewijn 1998 for the original term. The EyeLink is good for a video-based eye-tracker, but only marginally better than the SMI HiSpeed or the Tobii Spectrum. All these three VOGs have decent precision and accuracy, but they have a large number of artefacts that have been reported in literature: The accuracy errors due to varying pupil dilation, the PSOs, the erroneous measurement of saccade amplitudes below 1.5°, and of course artefacts due to movement of the participant. I do not mind using the EyeLink as baseline in a comparison, but it must be done with full acknowledgment of its weaknesses.

The proposed battery of tests is the first I have seen. I think the considerations here constitute the real merits of this paper. I have several comments, though:

Figure 4: Doubling the sampling frequency by interlacing left and right signals is also done in the LC Technology eye-tracker and in the Tobii Glasses 2. The Tobii glasses have the same problem as shown in C.

Task 6 the Pupil dilation task: When you were making this manipulation, why did you not check whether the accuracy of the gaze position is affected by the altered pupil dilation, like Jan Drewes and others have done? I think it should be part of this task.

Why do you not make a PSD-analysis (Power Spectral Density) analysis of the fixation task? A peak at 90Hz could indicate that the system is good enough to measure tremor. Ko, Poletti and Rucci 2016 made that comparison on the coil vs DPI.

Task 4: I showed in my talk at the Scandinavian Workshop on Applied Eye-Tracking 2018 that the EyeLink (and 10 other VOGs) mis-measure the amplitudes of small saccades (including micro-saccades). The paper is in review. I find your co-measurement very interesting, and I predict that if you make a item-by-item comparison between EyeLink vs Pupil Lab microsaccade amplitudes, you will find a large variation, as the two mis-measure different microsaccades in different directions (which I take your figure 10 C and D to confirm?).

lines 531-534: Yes, the relationship between sampling frequency and RMS is complex. However, RMS can be compared between systems of different sampling frequencies. What we used to think was an effect of sampling frequency was in fact an interaction between downsampling and the colour of noise (which changes with filtering). This was my talk at ECEM 2017. The paper is in review. My point is that you can exclude sampling frequency as an explanation. The binocular interlacing must be really bad for RMS and is likely the main explanation.

Note that the EyeLink is filtered by default (unless you turned off filters). It can be considered unfair to compare precision between a filtering and a non-filtering eye-tracker. Filters reduce RMS, but have no large effect on STD, which often confuses comparisons between eye-trackers, which find, just like you, that the two measures seem to tell a different story.

Task 8/9: I am quite confused that you compare the effect of head movements on eye movement data in a head mounted eye-tracker vs a remote system. From the view-point of the eye-camera in each system, the same head yaw or roll movement looks very different. It is not really a fair comparison, as you are not testing the same thing in both eye-trackers.

Line 721-729: A major difference between your study and Holmqvist 2017 appears to be that while you made conditions ideal, Holmqvist 2017 recorded with many sub-optimal participants wearing all kinds of glasses, lenses and make-up. Noone was rejected. This is again relevant on line 869.

Line 732: Wrong citation. You should cite Orquin and Holmqvist 2018

Line 783-788: It is very optimistic to think that Pupil Labs can be used to study microsaccades. I think your Figure 10 C and D hint that for measurement of small saccades, both eye-trackers have issues (which my SWAET talk from 2018 also shows).

Line 794: Why do you think that head roll is less common in mobile settings? I would think the opposite is true.

Line 809: Incomplete sentence about the pupil

Line 816: I think your test battery is a good one. But have you really proven reliability? And I do not think the battery is exhaustive. I know of several tests that you have not included (some referred to above, such as the effect of pupil dilation on gaze accuracy).

---

## Round 0.2 · Minor Revisions

Both reviewers were pleased with the revision and there are few minor comments pending. Once those are addressed, I will be happy to accept the paper.

Reviewer 1 ·

Basic reporting

See general comments

Experimental design

See general comments

Validity of the findings

See general comments

Additional comments

The authors have addressed all of my concerns. I have no further objections. I'd be happy to see this paper in print.

One thing that might be amended before publication is that the abstract and Figure 3 still refer to "detection" whereas the rest of the manuscript now states "classification", but this a minor detail and I trust the authors to catch this in e.g. the proofs.

·

Basic reporting

This updated version of the manuscript leaves very little to desire, and I am happy to recommend publication if the authors make a few minor additions and corrections.

Experimental design

I am happy with this version.

Validity of the findings

I am happy with this version.

Additional comments

My minor suggestions:

1) On page 19, figure 9, there are histograms that compare saccade amplitudes and fixation durations between the two eye-trackers (9F and 9G). This is a very coarse method for comparison of events. Recent literature, for instance

Zemblys, R., Niehorster, D. C., & Holmqvist, K. (2018). gazeNet: End-to-end eye-movement event detection with deep neural networks. Behavior research methods, 1-25.

discuss several recent methods for event-on-event comparisons between algorithms. I do not expect you to carry out the analysis in Zemblys et al, but it would be valuable to point out that these methods exist.

2) On page 28, line 828, there is a missing citation, to what I think might be a paper I have in review - entitled "Small eye-movements cannot be reliably measured by current video-based P-CR eye-trackers". I would recommend you to keep the sentence questioning the assumption that the EyeLink can measure microsaccades adequately, which my paper show that it cannot. You may not want to cite a paper in review, though.

---

## Round 0.3 · accepted · Accept

The extensive test set you describe in your paper will support eye tracking researchers make informed choices about which system to use for their paradigms.